# SpaCE-10: A Comprehensive Benchmark for Multimodal Large Language Models in Compositional Spatial Intelligence

**Ziyang Gong**[1*], **Wenhao Li**[2*], **Oliver Ma**[3*], **Songyuan Li**[4], **Zhaokai Wang**[1,3], **Songze Li**[3,5], **Jiayi Ji**[2,6], **Xue Yang**[1†], **Gen Luo**[2†], **Junchi Yan**[1], **Rongrong Ji**[2]
[1]SCS & SAIS & SAI, Shanghai Jiao Tong University
[2]Xiamen University [3]Shanghai Artifical Intelligence Laboratory
[4]Sun Yat-sen University [5]Fudan University [6]National University of Singapore
Code:https://github.com/VisionXLab/SpaCE-10

## Abstract

Multimodal Large Language Models (MLLMs) have achieved remarkable progress in various multimodal tasks. To pursue higher intelligence in space, MLLMs require integrating multiple spatial capabilities, even for handling simple and normal tasks. However, existing benchmarks struggle to comprehensively evaluate the spatial intelligence of common MLLMs from the atomic level to the compositional level. To fill this gap, we present SpaCE-10, a comprehensive benchmark for compositional spatial evaluations. In SpaCE-10, we define 10 atomic spatial capabilities, which are combined to form 8 compositional capabilities. Based on these definitions, we propose a novel hierarchical annotation pipeline to generate high-quality and diverse question-answer (QA) pairs. With over 150+ hours of human expert effort, we obtain over $5k$ QA pairs for 811 real indoor scenes in SpaCE-10, which covers various evaluation settings like point cloud input and multi-choice QA. We conduct an extensive evaluation of common MLLMs on SpaCE-10 and find that even the most advanced MLLM still lags behind humans by large margins. Through our careful study, we also draw several significant findings that benefit the MLLM community. For example, we reveal that the shortcoming of counting capability greatly limits the compositional spatial capabilities of existing MLLMs. The Evaluation code has been released.

## 1 Introduction

Recent years have witnessed the rapid development of multimodal large language models (MLLMs) (OpenAI, 2025; Bai et al., 2025; Wang et al., 2025), which continually narrows the gap between machines and humans in multimodal tasks (Liu et al., 2024). The significant progress has motivated researchers to pursue higher machine intelligence in the real world (Huang et al., 2023; Hong et al., 2023; Luo et al., 2025). Focus on the scene in Fig. 1, imagine you are about to head out the door and tell your home robot, *'I forgot my watch, please bring it to me. I remember it's near the nightstand.'* To succeed, the robot must know what a watch is, plan to the nightstand, reason about spatial relations near, localize the watch among distractors, retrieve it, and return under diverse changing viewpoints. Solving this 'simple, everyday task' requires the on-the-fly composition of a diverse set of spatial capabilities. This raises a central question: *Can current MLLMs master these spatial capabilities and compose them seamlessly in real-world scenarios?*

While existing benchmarks have made valuable explorations of the spatial intelligence of multimodal large language models (MLLMs) (Yang et al., 2024; Ma et al., 2024; Linghu et al., 2024; Yang et al., 2025b; Jia et al., 2025), they seldom make these capabilities explicit or design tasks that systematically combine them. Early benchmarks (Azuma et al., 2022; Ma et al., 2022; Yan et al., 2023; Ye et al., 2022; Chen et al., 2020) mainly focus on the assessment of less-combined

---

*Equal Contribution
†Correspondence authors

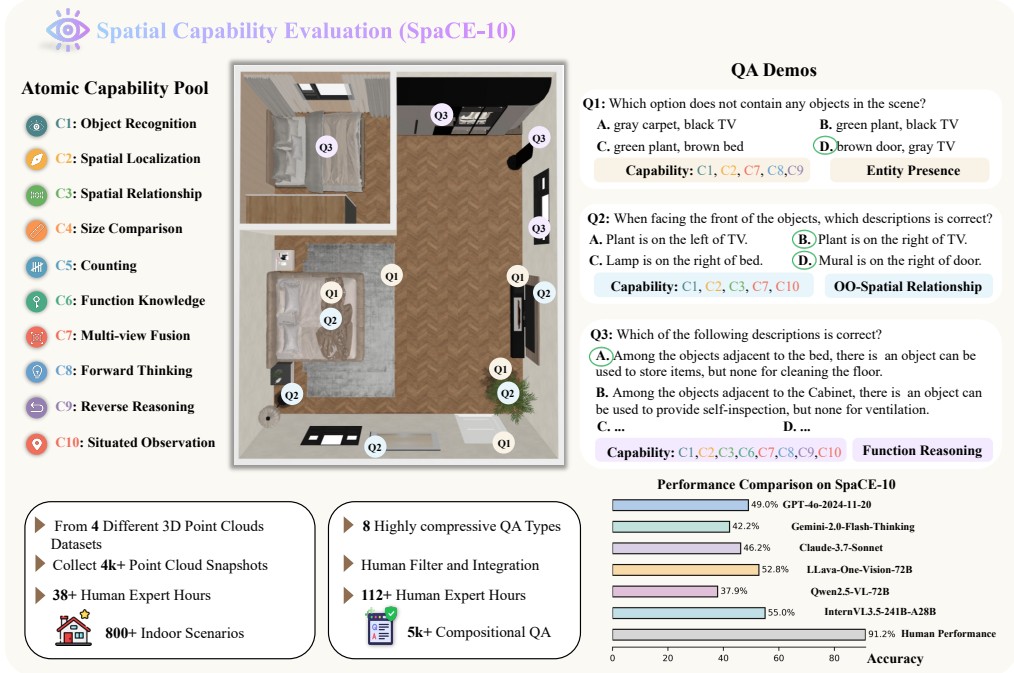

Figure 1: **Overview of SpaCE-10 benchmark.** SpaCE-10 takes over 150 human expert hours to collect 5k+ QA pairs in 811 indoor scenes, which can evaluate MLLMs from 10 atomic capabilities to 8 compositional capabilities. Through evaluations, SpaCE-10 indicates that even the most advanced MLLM still lags far behind humans by large margins. Green cirle means the correct answer.

capabilities like object recognition and spatial localization, while compositional ones still remain to be defined and evaluated. Recent benchmarks (Yang et al., 2024; 2025b; Jia et al., 2025) aim to evaluate the spatial intelligence of MLLMs through more compositional questions, but still fail to reflect the role of different spatial capabilities in compositional reasoning. More importantly, existing spatial benchmarks struggle to satisfy the evaluation needs of existing MLLMs in terms of scenes, modalities, and question types, *etc*. As shown in Tab. 1, the number of scenarios in existing benchmarks is usually less than 400, which makes it difficult to cover various practical situations.

To fill these gaps, this paper proposes SpaCE-10 (Spatial Capability Evaluation), a capability-focus question-answer (QA) benchmark with an atomic capability pool (Fig. 1). This pool highlights the 10 core spatial capabilities (C1-C10) for MLLMs in real-world deployment. In SpaCE-10, there are 8 meticulously designed and systematically combined QA types, each of which covers more than 5 atomic capabilities. Hence, SpaCE-10 can not only assess the Compositional Spatial Intelligence (CSI) of MLLMs but can also reflect the impact of different atomic capabilities in spatial comprehension.

Based on this design principle, we propose an innovative hierarchical annotation pipeline in SpaCE-10. Specifically, we collect over 800 real indoor scanned scenes from four public datasets. For each scene, we present an automated pipeline to generate structured data that can describe different types of information in the scene, *e.g.,* appearance and relationship. Based on this information, a multi-stage semi-automated pipeline is adopted to generate basic QA pairs, conduct quality verification, and perform the capability integration. Our SpaCE-10 consists of more than 5,000 high-quality QA pairs, covering various settings of existing MLLMs, *e.g.,* point cloud input and multi-choice question types. As shown in Tab. 1, SpaCE-10 demonstrates greater diversity than previous benchmarks in data distribution, annotation process, and evaluation settings, showing promising all-around evaluation ability for compositional spatial intelligence.

We conduct extensive and systematic evaluations of mainstream MLLMs on SpaCE-10, including 4 close-source MLLMs and nearly **50** open-source MLLMs ranging from 1B to 241B. Experimental results show that even the most advanced MLLMs are still far behind humans in compositional spatial intelligence, *i.e.,* 53.1% of GPT-5 *vs.* 91.2% of human. Meanwhile, 2D MLLMs demonstrate

Table 1: **Comparison of SpaCE-10 with existing spatial benchmarks.** Our SpaCE-10 contains the most diverse scenarios and QA types, covering various evaluation settings of existing MLLMs. CSI means the investigation of Compositional Spatial Intelligence. SCN, OBJ, HM3D, 3RS, ARK represent ScanNet (Dai et al., 2017), Objaverse, Habitat-Matterport-3D (Ramakrishnan et al., 2021), 3RScan (Wald et al., 2019), and ArkitScene (Baruch et al., 2021). Sim. denotes similarity-based metrics. 2D&3D means the benchmarks support both 2D imagery and 3D point cloud-based MLLMs.

| Dataset | Scenario Source | Scene | Q&A | Metric | 2D & 3D | Multi-Answer | CSI |
|---|---|---|---|---|---|---|---|
| 3DQA (Zhao et al., 2022) | SCN | - | 902 | Sim. | × | × | × |
| ScanQA (Azuma et al., 2022) | SCN | 167 | 10k | Sim. | ✓ | × | × |
| FE-3DGQA (Zhao et al., 2022) | SCN | 100 | 3.9k | Sim. | × | × | × |
| SQA3D (Ma et al., 2022) | SCN | 132 | 6.9k | Sim. | ✓ | × | × |
| CLEVER3D (Yan et al., 2023) | SCN | 133 | 10k | Sim. | × | × | × |
| 3D-LLM (Hong et al., 2023) | OBJ,SCN,HM3D | - | 30k | Sim. | × | × | × |
| M3DBench (Li et al., 2023) | SCN | - | 1.5k | Sim.+LLM | × | × | × |
| MSQA (Linghu et al., 2024) | SCN,3RS,ARK | 381 | 3.5k | LLM | × | × | × |
| VSI (Yang et al., 2024) | SCN,3RS,SCN++ | 288 | 5.0k | Acc. | × | × | × |
| **SpaCE-10 (Ours)** | **SCN, 3RS, ARK, SCN++** | **811** | **5.0k** | **Acc.** | ✓ | ✓ | ✓ |

much stronger capabilities than 3D MLLMs on SpaCE-10, showing great potential for image-based spatial reasoning. In addition, existing MLLMs greatly fall short in multiple-answer QAs, suggesting their inferior complex reasoning abilities. Our further study also reveals that the shortcoming of counting capability greatly limits the compositional spatial capabilities of existing MLLMs. These findings provide valuable directions for the community to develop more capable MLLMs in terms of spatial intelligence. Overall, our main contributions are summarized as follows:

- We present SpaCE-10, a comprehensive benchmark for compositional spatial intelligence. SpaCE-10 is the most diverse benchmark that can assess the capabilities of MLLMs from the atomic level to the compositional level. It also covers various evaluation settings, including 3D inputs and multi-choice questions.
- We propose an innovative hierarchical annotation pipeline in SpaCE-10, which first produces structured descriptions of scenes via an automated pipeline and then generates compositional QA pairs through a multi-stage semi-automated pipeline. The hierarchical pipeline ensures the quality, diversity, and controllability of the generated QA pairs.
- We conduct extensive evaluations for nearly 50 open- and close-source MLLMs on SpaCE-10 Through our in-depth analysis, we draw several significant findings that will benefit the spatial intelligence of future MLLMs in the community.

## 2 RELATED WORK

Early works start spatial intelligence mainly with two directions: (i) 2D abstract reasoning with logic-puzzle and geometric-panel tasks (e.g., Raven-style matrices) (Gonthier, 2022; Xiao et al., 2024; Xu et al., 2025; Ramakrishnan et al., 2024), and (ii) simplified images with only a few objects, where queries test basic relations such as above/below/left/right/size (Tong et al., 2024b; Kamath et al., 2023; Bagherinezhad et al., 2016). As attention shifts to realistic environments, 3D scene benchmarks (Azuma et al., 2022; Ma et al., 2022; Linghu et al., 2024; Ye et al., 2022; Li et al., 2023) emerge and expand to richer tasks, such as route planning and situated perception from specified viewpoints. However, they typically adopt point-cloud inputs and still treat spatial ability as a single block. With the rise of MLLMs, newer studies (Ma et al., 2024; Yang et al., 2024; 2025b; Jia et al., 2025; He et al., 2025) examine 3D spatial comprehension directly from 2D images or videos. Yet most works do not make capabilities explicit or delve into them. To this end, SpaCE-10 defines an atomic capability pool (C1-C10) across perception and reasoning to assess a model's CSI. It also supports both 2D images and 3D point clouds, offering a new perspective for advancing spatial intelligence in MLLMs.

# 3 SpaCE-10

## 3.1 Overview

**Construction** The images in SpaCE-10 are from four 3D indoor-scene scan datasets, including ScanNet++ , ScanNet (SCN), 3RScan (3RS), and ARkitScene (ARK) with over 800 real indoor scenes, including a wide variety of environments such as living rooms, classrooms, bathrooms, kitchens, and more. Finally, SpaCE-10 consists of 8 QA types that are EQ (Entity Quantification), SQ (Scene Quantification), SA (Size Assessment), OO (Object-Object Spatial Relationship), OS (Object-Scene Spatial Relationship), EP (Entity Presence), FR (Functional Reasoning), and SP (Spatial Planning). Also it includes 10 atomic spatial capabilities of C1 (Object Recognition), C2 (Spatial Localization), C3 (Spatial Relationship), C4 (Size Comparison), C5 (Counting), C6 (Funciton Knowledge), C7 (Multi-view Fusion), C8 (Forward Thinking), C9 (Reverse Reasoning), and C10 (Situated Observation). Definition of QA and capabilities are in Sec. B and C, respectively.

**Analysis** In Fig. 2 (a), we show the numbers of each type of QA in SpaCE-10. Among them, blue ones belong to Perception, and purple ones belong to Reasoning. In (b) and (C), we demonstrate the average vocabulary size (unique word number) and character length of the Question and Option of 6 QA types. EQ and SQ are excluded because their options are numbers.

For Capability, in (d) we show the number of capability contained in each QA type. (e) represents the capability coverage in each QA type, revealing a three-tier hierarchy. (1) **Foundation**: C1 (Object Recognition), C7 (Multi-view Fusion), C8 (Forward Thinking) appear in 100% of QA types since they are prerequisites for almost every task. (2) **Bridge**: C2 (Spatial Localization) covers 87% and links both 'quantification' and 'relation/plan' QA types. (3) **Specialist**: C3 (Spatial Relationship) and C10 (Situated Observation) are mid-frequency (45%), activated when viewpoint-dependent relations matter; C9 (Reverse Reasoning) attaches to causal/plan problems; C4 (Size Comparison), C5 (Counting), and C6 (Function Knowledge) appear in specific QA types. (f) shows the co-occurrence of each capability. Beyond of (d), we observe the tightest pair is C3 (Spatial Relationship) with C10 (Situated Observation) = 4/8 (OO, OS, FR, SP), which means they appear together in 3 QA types, showing that when relative relations are asked, a specified viewpoint is usually required. Moreover, C9 (Reverse Reasoning) appears mainly with planning-style tasks: C2 (Spatial Localization) with C9 (Reverse Reasoning) = 3/8 (EP, FR, SP), while C3 (Spatial Relationship) with C9 (Reverse Reasoning) = 2/8 (FR, SP) and C10 (Situated Observation) with C9 (Reverse Reasoning) = 2/8 (FR, SP). The design motivation of the atomic capability pool can be found at Sec. E.

## 3.2 Hierarchical Annotation Pipeline

**Overview.** As shown in Fig 3 (a), our annotation pipeline consists of 5 stages from data preparation to high-quality QA generation. In stage 1, we employ 3 human experts to manually collect snapshots of 3D point cloud scans from 4 to 6 different directions, with over **38 hours** of human expertise to maintain the high quality. In stage 2, we combine the collected snapshots and video frames to generate the structural data that describes different aspects of information in the scene. In stage 3, we leverage GPT-4o to generate over 10k Basic QA covering atomic capabilities with former structural data. In stage 4, human experts again manually filter the low-quality QA pairs, costing over **112 hours** of 3 experts and resulting in over 8k+ QA pairs. Finally, in stage 5, we design a template-based strategy to integrate the spatial capabilities in QA types, yielding the final QA pairs. The ablation study on the effectiveness of the annotation pipeline can be found at the Sec. F.

**Structural Data Generation.** As shown in Fig. 3 (b), this pipeline follows a progressive design to generate structural data with 6 steps: (1) Initially, 10 keyframes are selected from the video of each scenario by combining the CLIP vision encoder (Radford et al., 2021) and the k-means algorithm. (2) Based on 2D keyframes, we leverage GPT-4o to generate a 2D caption for each scene, which covers information of appearance, size, and spatial relationships. (3) We reuse GPT-4o as an inspector to refine the 2D captions by removing incorrect and redundant information. (4) The manually collected 3D snapshots will be combined with keyframes for 3D caption generation. These high-quality snapshots contain rich global information about the whole scenes, which can provide considerable scene-level spatial information. (5) The inspector again checks and refines the 3D caption. (6) Finally, the rule-based extractor will be applied to obtain structural data for the following QA generation.

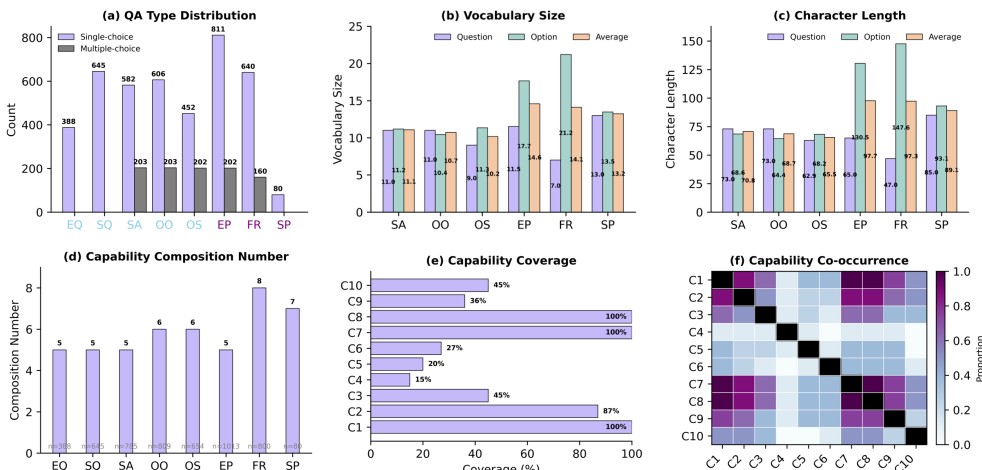

Figure 2: **Dataset analysis of SpaCE-10.** (a) Number distribution of each QA type. SpaCE-10 consists of 8 QA types that are EQ (Entity Quantification), SQ (Scene Quantification), SA (Size Assessment), OO (Object-Object Spatial Relationship), OS (Object-Scene Spatial Relationship), EP (Entity Presence), FR (Functional Reasoning), and SP (Spatial Planning). (b) Average vocabulary size per QA type for question, option, and average. (c) Average character length per QA type. (d) Coverage of the atomic capabilities (C1-C10). (e) The correlation between human expert accuracy and average character length across six QA types. (f) Capability co-occurrence matrix.

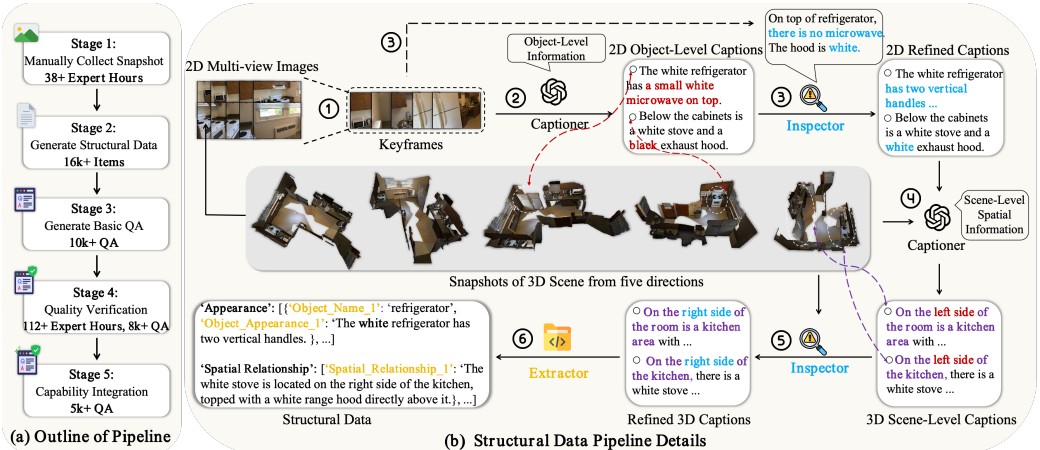

Figure 3: **Illustration of our hierarchical annotation pipeline.** We generate structural data to construct over 10$k$ QA pairs, and performs capability integration to obtain over 5$k$ QA pairs with 10 compositional capabilities. This process takes over 150 expert hours for data collection and filtering.

**QA Generation.** For QA generation, we adopt 3 approaches: template-based, MLLM-based, and human-based generation. For EQ, we use template-based method, and SP is manually designed by 2 human expert. For rest QA types, we leverage GPT-4o to generate. Notably, as we mentioned earlier, the questions in SpaCE-10 are composed of multiple atomic capabilities. However, such highly integrated questions are difficult for current MLLMs to generate directly. Therefore, for SA, OO, OS, EP, FR five QA types, we propose to first generate a basic version of QA, namely basic QA, and then enhance its embedded capabilities. The details of QA generation is in Sec. D.

**Cross-Capability Integration Strategy.** We apply three strategies to integrate cross-capabilities: (1) For SA, OO, and OS, we integrate an additional C7 (Multi-view Fusion). In the original setting, the four options usually refer to the same object (or object pair). We regroup multiple same-type, same-scene QAs into a single QA so that each option points to different objects. This forces MLLMs to search across the entire scene to find all mentioned entities, enabling a more holistic spatial perception. (2) For EP, we add C7 and C9 (Reverse Reasoning), expanding the question to involve multiple objects. We then reverse the question type from 'which object exists' to 'which object

does not exist', integrating reverse reasoning capabilities. (3) For FR, we add C7, C9, and C10 (Situated Observation). The basic FR question asks 'which option correctly describes the function of an object near *central object*'. The four options are structured as 'Object' which can be used to 'Function'. In the integrated version, the question is simply revised to: 'Which of the following is correct?'. But Each option involves a different central object, with the structure: 'Among the objects adjacent to *central object*, there is one can be used to 'Function', but lacks two objects can be used to 'Function'. This change prevents potential leakage of prior knowledge in options, such as object names, and places greater emphasis on the model's understanding of functional roles. The examples are in the Sec. D.2.

## 3.3 QUALITY VERIFICATION

For the quality verification process, we rely on manual filtering by 3 human experts. We set up an user interface for validation and employed two human experts to perform the evaluation. In this process, the evaluation criteria include checking for incorrect options, invalid answers, missing data, or questions involving objects not present in the snapshots. The low-quality data will be directly deleted. This quality control process takes over 112 hours to check and filter low-quality QA pairs. By employing human validation, we ensure that only high-quality and contextually accurate questions are retained for the final benchmark. Related visualizations are attached in the Sec. H.

## 4 EXPERIMENTS

### 4.1 SETUP

In our experiments, we test nearly 50 close-source and open-source MLLMs on SpaCE-10, including 3D MLLMs LEO and GPT4Scene, GPT-5, InternVL3.5, and so on. During the evaluation, except for Leo, which is evaluated by their own framework, other MLLMs are evaluated by the LMMs-Eval (Zhang et al., 2024) with 8 frames as input, and open-source MLLMs are tested on Nvidia A100 GPUs. For the response and answer alignment, we follow the prompt of MMBench (Liu et al., 2024) and use GPT-4o-2024-11-20 for the judgment. Notably, we removed the random assignment in LMMs-Eval, so the model's performance is likely to be lower than the random baseline (25%).

### 4.2 OVERALL RESULTS

**Human *vs.* MLLMs.** We first compare human and MLLMs' performance on SpaCE-10. The 'Human' score is taken from the average score of 6 human experts. The results in Tab. 2 indicate that although human performance does not meet expectations when facing with more complex numerical and reasoning tasks, the total score of 91.2% is still significantly higher than all existing MLLMs. In comparison, the best open-source MLLM only achieves an average score of 55.0%, and the best close-source models only achieve a score of 53.1%. These results demonstrate that the compositional spatial intelligence of MLLMs is far below the human level.

**3D MLLMs *vs.* 2D MLLMs.** In Tab. 2, we evaluate LEO and GPT4Scene as representatives of 3D-related MLLMs. Notably, input to the LEO model requires point clouds of the objects relevant to the questions. To ensure a fair comparison, we adjust the input for LEO by randomly sampling 1024 points from the entire scene's point clouds. The results show that LEO scored 11.1% overall in SpaCE-10, which is significantly lower than GPT4Scene (34.5%). Compared to MLLMs with scale $\leq$ 7B, the performance of LEO-7B is also substantially lower. We argue that one of the limitations of current 3D MLLMs is that they are designed to focus on specific objects and have difficulty processing the entire scene's point clouds as input. Additionally, they likely sacrifice multimodal conversational abilities for understanding scans. These results also indicate that 2D MLLMs have greater potential in visual spatial intelligence comprehension than 3D MLLMs. **Open-Source *vs.* Close-Source.** In close-souce MLLMs experiments, GPT-5 achieves the best overall performance, ranking 8th with a score of 53.2%. It excels in perception tasks, such as in SA (Size Assessment) with 69.7%, OO (Object-Object Spatial Relationship) with 60.7%, and FR (Functional Reasoning) with 66.8%, indicating strong accuracy in recognizing size and position. Additionally, GPT-4o achieves the highest score in EQ (Entity Quantification) among all tested models, with a score of 58.3%. However, GPT-4o struggles with SP (Spatial Planning), where it scores the lowest among all

Table 2: **Single-answer performance ranking of nearly 50 MLLMs on SpaCE-10 benchmark.**

| Models | Rank | Perception | | | | | Reasoning | | | Overall |
|---|---|---|---|---|---|---|---|---|---|---|
| | | EQ | SQ | SA | OO | OS | EP | FR | SP | |
| Human | 1 | 91.3 | 88.5 | 90.2 | 93.4 | 95.6 | 91.1 | 90.3 | 86.3 | 91.2 |
| **3D MLLMs** | | | | | | | | | | |
| LEO-7B (Huang et al., 2023) | 47 | 15.8 | 0.0 | 16.7 | 16.5 | 25.2 | 5.5 | 5.7 | 13.3 | 11.1 |
| GPT4Scene-7B (Qi et al., 2025) | 36 | 30.9 | 37.7 | 38.0 | 38.9 | 41.6 | 29.5 | 28.0 | 32.5 | 34.5 |
| **Close Source 2D MLLMs** | | | | | | | | | | |
| GPT-5 (OpenAI, 2025) | 3 | 42.0 | 43.0 | 71.0 | 60.7 | 36.5 | 50.3 | 66.8 | 36.0 | 53.4 |
| GPT-4o-2024-11-20 (Achiam et al., 2023) | 9 | 58.3 | 32.8 | 56.2 | 58.3 | 56.2 | 41.6 | 52.2 | 23.7 | 49.0 |
| Gemini-2.0-Flash-Thinking (Team et al., 2023) | 20 | 34.3 | 25.6 | 53.1 | 42.6 | 53.8 | 42.2 | 46.7 | 31.2 | 42.2 |
| Claude-3.7-Sonnet (cla) | 14 | 46.0 | 44.3 | 49.1 | 46.0 | 49.1 | 44.3 | 49.3 | 25.0 | 46.2 |
| **Open Source 2D MLLMs** | | | | | | | | | | |
| ▼ *Scale < 7B* | | | | | | | | | | |
| InternVL2.5-1B (Chen et al., 2024) | 33 | 33.0 | 54.1 | 18.8 | 43.6 | 29.9 | 26.7 | 41.0 | 23.7 | 35.3 |
| InternVL3-1B (Zhu et al., 2025) | 24 | 30.7 | 55.7 | 27.9 | 44.6 | 31.6 | 47.8 | 41.9 | 30.0 | 41.4 |
| InternVL3.5-1B (Wang et al., 2025) | 38 | 34.8 | 41.7 | 29.4 | 42.7 | 25.9 | 21.9 | 40.2 | 33.8 | 33.5 |
| InternVL2.5-2B (Chen et al., 2024) | 42 | 32.2 | 26.8 | 27.0 | 36.6 | 28.8 | 21.7 | 48.2 | 36.2 | 31.4 |
| InternVL3-2B (Zhu et al., 2025) | 17 | 41.5 | 45.9 | 45.4 | 45.7 | 31.9 | 45.7 | 48.7 | 41.3 | 44.2 |
| InternVL3.5-2B (Wang et al., 2025) | 35 | 35.6 | 28.4 | 42.2 | 45.7 | 32.3 | 20.1 | 45.8 | 20.0 | 34.6 |
| Qwen2.5-VL-3B-Instruct (Bai et al., 2025) | 34 | 31.7 | 23.3 | 47.1 | 51.7 | 31.6 | 25.5 | 37.0 | 21.2 | 34.8 |
| SpaceOm◇ | 40 | 21.8 | 24.5 | 47.3 | 49.7 | 32.7 | 21.9 | 36.7 | 25.0 | 33.2 |
| SpaceQwen◇ | 32 | 31.2 | 26.1 | 41.2 | 52.3 | 35.2 | 28.4 | 36.4 | 22.5 | 35.4 |
| SpaceThinker◇ | 37 | 32.7 | 22.4 | 46.7 | 50.5 | 33.4 | 22.4 | 36.9 | 24.2 | 34.1 |
| VILA1.5-3B (Lin et al., 2024) | 45 | 25.0 | 9.1 | 31.7 | 34.6 | 31.6 | 35.3 | 12.9 | 33.7 | 26.1 |
| InternVL2.5-4B (Chen et al., 2024) | 29 | 34.3 | 23.4 | 50.2 | 50.8 | 16.2 | 21.7 | 56.0 | 33.7 | 35.9 |
| MiniCPM-v4-4B (Yao et al., 2025) | 27 | 38.1 | 32.7 | 41.1 | 49.0 | 36.5 | 29.3 | 50.0 | 30.0 | 39.0 |
| InternVL3.5-4B (Wang et al., 2025) | 30 | 38.9 | 12.9 | 48.7 | 50.7 | 27.9 | 33.9 | 37.0 | 35.0 | 35.5 |
| ▼ *Scale ≤ 14B* | | | | | | | | | | |
| Qwen2.5-VL-7B-Instruct (Bai et al., 2025) | 39 | 32.7 | 36.9 | 36.9 | 35.3 | 32.3 | 27.6 | 34.2 | 27.5 | 33.3 |
| LLaVA-v1.5-7B (Liu et al., 2023) | 43 | 31.2 | 31.3 | 30.5 | 35.7 | 22.9 | 10.7 | 57.4 | 32.5 | 30.7 |
| LLaVA-OneVision-7B (Li et al., 2024) | 15 | 37.4 | 33.8 | 46.4 | 57.3 | 34.5 | 43.3 | 61.6 | 21.2 | 45.2 |
| MiMo-VL-RL-8B (Xiaomi, 2025) | 31 | 23.7 | 35.0 | 46.4 | 41.3 | 34.7 | 32.2 | 32.5 | 36.1 | 35.5 |
| Cambrian-8B (Tong et al., 2024a) | 44 | 22.6 | 18.6 | 34.8 | 32.6 | 32.3 | 25.1 | 41.4 | 23.7 | 29.5 |
| VILA1.5-8B (Lin et al., 2024) | 46 | 25.7 | 8.2 | 27.5 | 32.7 | 17.2 | 12.4 | 26.7 | 23.7 | 20.9 |
| InternVL2.5-8B (Chen et al., 2024) | 21 | 33.2 | 36.0 | 50.0 | 55.0 | 33.6 | 27.1 | 59.1 | 32.5 | 41.8 |
| InternVL3-8B (Zhu et al., 2025) | 25 | 36.6 | 29.5 | 42.9 | 51.7 | 34.5 | 26.6 | 60.6 | 37.5 | 40.0 |
| InternVL3.5-8B (Wang et al., 2025) | 26 | 37.1 | 28.5 | 61.7 | 49.8 | 35.4 | 17.6 | 54.8 | 36.3 | 39.5 |
| Gemma3-12B (Team et al., 2025a) | 22 | 41.8 | 42.1 | 55.1 | 46.5 | 35.6 | 25.0 | 53.2 | 27.5 | 41.5 |
| InternVL3-14B (Zhu et al., 2025) | 12 | 39.7 | 28.7 | 54.4 | 58.1 | 38.1 | 51.3 | 56.6 | 35.0 | 47.3 |
| InternVL3.5-14B (Wang et al., 2025) | 10 | 41.0 | 47.6 | 65.3 | 52.1 | 34.5 | 45.4 | 54.3 | 30.0 | 48.8 |
| ▼ *14B<Scale< 72B* | | | | | | | | | | |
| InternVL3.5-20B-A4B (Wang et al., 2025) | 8 | 37.4 | 43.1 | 64.1 | 58.7 | 41.4 | 54.1 | 57.6 | 28.8 | 51.6 |
| InternVL2.5-26B (Chen et al., 2024) | 19 | 34.3 | 29.3 | 62.6 | 55.4 | 33.0 | 29.2 | 61.8 | 33.7 | 43.3 |
| Gemma3-27B (Team et al., 2025a) | 23 | 39.4 | 21.7 | 63.5 | 48.5 | 37.8 | 33.2 | 51.5 | 30.0 | 41.5 |
| Qwen2.5-VL-32B-Instruct (Bai et al., 2025) | 41 | 19.9 | 26.5 | 48.9 | 36.8 | 32.3 | 31.1 | 30.1 | 32.5 | 32.6 |
| InternVL2.5-38B (Chen et al., 2024) | 16 | 38.1 | 36.1 | 64.4 | 54.3 | 36.8 | 27.4 | 63.0 | 37.5 | 45.1 |
| InternVL3-38B (Zhu et al., 2025) | 4 | 36.3 | 41.6 | 69.5 | 60.1 | 36.3 | 58.6 | 60.8 | 35.0 | 53.1 |
| InternVL3.5-38B (Wang et al., 2025) | 18 | 42.3 | 28.4 | 62.8 | 59.1 | 37.6 | 25.4 | 59.8 | 28.8 | 43.9 |
| ▼ *Scale ≥ 72B* | | | | | | | | | | |
| GLM-4.5V (Team et al., 2025b) | 7 | 38.9 | 41.1 | 65.5 | 61.1 | 36.7 | 61.2 | 49.3 | 31.3 | 51.6 |
| LLaVA-OneVision-72B (Li et al., 2024) | 5 | 44.1 | 38.3 | 67.9 | 64.5 | 40.3 | 46.7 | 67.3 | 36.2 | 52.8 |
| Qwen2.5-VL-72B-Instruct (Bai et al., 2025) | 28 | 32.4 | 34.9 | 55.7 | 40.9 | 32.1 | 36.5 | 38.0 | 33.7 | 38.7 |
| InternVL2.5-78B (Chen et al., 2024) | 13 | 27.8 | 45.0 | 62.4 | 64.4 | 40.3 | 23.7 | 67.3 | 40.0 | 47.1 |
| InternVL3-78B (Zhu et al., 2025) | 6 | 36.8 | 48.2 | 65.3 | 61.6 | 43.8 | 44.4 | 64.3 | 46.3 | 52.5 |
| Qwen3-VL-235B-A22B-Instruct | 11 | 37.3 | 29.6 | 66.4 | 62.9 | 37.8 | 38.6 | 64.0 | 26.8 | 47.9 |
| InternVL3.5-241B-A28B (Wang et al., 2025) | 2 | 35.8 | 39.1 | 68.2 | 63.5 | 46.2 | 64.2 | 58.6 | 40.0 | 55.0 |

◇ Models proposed by RemyxAI (SpaceVLMs series, https://huggingface.co/remyxai/SpaceQwen2.5-VL-3B-Instruct).

tasks, suggesting a limitation in scene-level planning ability. In open-source MLLMs, among models with a scale of over 72B, InternVL3.5-241B-A28B delivers outstanding performance, ranking first overall with a score of 55.0%. It outperforms other models in nearly all tasks. These results suggest that the gap between open-source and close-source models has been significantly narrowed, and some open-source MLLMs even outperform close-source models, especially in compositional spatial intelligence.

**Single-Answer *vs*. Multiple-Answer.** In this experiment, we choose two close MLLM series of similar performance to make a comparison on single-answer and multiple-answer questions. The results in Tab. 4 show that MLLMs perform significantly worse on multiple-answer tasks compared to single-answer tasks. Smaller models like InternVL2.5-1B and 2B score over 30.0% on single-answer tasks, while in multiple-answer tasks, their scores often fall below 5%, which is even worse than random selection. As the model size increases to 78B, MLLMs show greater robustness for different QA types. These results lead to an interesting preliminary conclusion: smaller models may overfit to the single-answer task format, while larger models seem to have learned more fundamental compositional spatial intelligence. However, the Qwen series exhibits an almost opposite trend, with

Table 3: **Performance of MLLMs on single-choice and multiple-choice QA pairs.** Results show that MLLMs, especially smaller ones, tend to overfit to single-choice questions.

| Models | Single-choice (3091) | | | | | Multiple-choice (970) | | | | | Overall ↑ | | Score ↑ |
|---|---|---|---|---|---|---|---|---|---|---|---|---|---|
| | SA | OO | OS | EP | FR | SA | OO | OS | EP | FR | Single | Multiple | multiple/single |
| *InternVL2.5 Series* | | | | | | | | | | | | | |
| InternVL2.5-1B (Chen et al., 2024) | 18.8 | 43.6 | 29.9 | 26.8 | 41.0 | 4.4 | 3.9 | 4.7 | 0.5 | 11.4 | 32.3 | 3.0 | 0.09 |
| InternVL2.5-2B (Chen et al., 2024) | 27.0 | 36.6 | 28.8 | 21.7 | 48.0 | 4.7 | 2.0 | 1.3 | 1.5 | 3.5 | 32.4 | 4.1 | 0.13 |
| InternVL2.5-8B (Chen et al., 2024) | 50.0 | 55.0 | 33.6 | 27.1 | 59.1 | 8.4 | 14.8 | 8.7 | 12.9 | 1.5 | 45.0 | 10.6 | 0.24 |
| InternVL2.5-38B (Chen et al., 2024) | 64.4 | 54.3 | 36.8 | 27.4 | 63.0 | 47.8 | 37.9 | 7.3 | 21.3 | 46.8 | 49.2 | 32.2 | 0.65 |
| InternVL2.5-78B (Chen et al., 2024) | 62.4 | 64.4 | 40.3 | 23.7 | 67.3 | 38.4 | 33.5 | 12.1 | 12.9 | 45.8 | 51.6 | 28.5 | 0.55 |
| *Qwen2.5-VL Series* | | | | | | | | | | | | | |
| Qwen2.5-VL-3B-Instruct (Bai et al., 2025) | 47.1 | 51.7 | 31.6 | 25.5 | 37.0 | 21.7 | 24.1 | 8.0 | 15.3 | 20.7 | 38.6 | 17.9 | 0.46 |
| Qwen2.5-VL-7B-Instruct (Bai et al., 2025) | 36.9 | 35.3 | 32.3 | 27.6 | 34.2 | 20.2 | 11.8 | 14.7 | 13.4 | 10.4 | 33.3 | 14.1 | 0.42 |
| Qwen2.5-VL-32B-Instruct (Bai et al., 2025) | 48.9 | 36.8 | 32.3 | 31.1 | 30.1 | 25.6 | 6.9 | 8.0 | 10.4 | 12.9 | 35.8 | 12.8 | 0.36 |
| Qwen2.5-VL-72B-Instruct (Bai et al., 2025) | 55.7 | 40.9 | 32.1 | 36.5 | 38.0 | 17.2 | 13.3 | 13.0 | 15.7 | 15.9 | 40.6 | 15.0 | 0.37 |
| *Others* | | | | | | | | | | | | | |
| Cambrian-8B (Tong et al., 2024a) | 34.8 | 32.6 | 32.3 | 25.1 | 41.4 | 5.9 | 12.8 | 5.9 | 0.5 | 20.6 | 33.2 | 9.1 | 0.27 |
| VILA1.5-8B (Lin et al., 2024) | 27.5 | 32.7 | 17.2 | 12.4 | 26.7 | 0.0 | 0.5 | 1.5 | 2.0 | 26.9 | 23.3 | 6.2 | 0.27 |
| LLaVA-OneVision-72B (Li et al., 2024) | 67.9 | 64.5 | 40.3 | 46.7 | 67.3 | 38.9 | 32.0 | 13.3 | 34.7 | 35.8 | 57.3 | 30.9 | 0.54 |

Table 4: **Accuracy comparison on basic and compositional QA pairs.** The results reveal the relationship between the performance and integration of capacities.

| Task | Integrated | Capability | InternVL2.5-1B | InternVL2.5-8B | Qwen2.5-VL-3B | Qwen2.5-VL-7B | Average-C(%) |
|---|---|---|---|---|---|---|---|
| **SA** | ✗ | C1,C2,C4,C8 | 37.8 | 66.4 | 60.0 | 64.8 | 57.3 |
| | ✓ | **+C7** | 18.8 (↓ 19.0) | 50.0 (↓ 16.4) | 47.1 (↓ 12.9) | 36.9 (↓ 27.9) | 38.2 (↓ 19.1) |
| **OO** | ✗ | C1,C2,C3,C8,C9,C10 | 52.0 | 66.4 | 56.4 | 41.2 | 54.0 |
| | ✓ | **+C7** | 43.6 (↓ 8.4) | 55.0 (↓ 11.4) | 51.7 (↓ 4.7) | 35.3 (↓ 5.9) | 46.4 (↓ 7.6) |
| **OS** | ✗ | C1,C2,C3,C9,C10 | 44.2 | 54.8 | 50.8 | 54.3 | 51.0 |
| | ✓ | **+C7** | 29.9 (↓ 21.6) | 33.6 (↓ 21.2) | 31.6 (↓ 19.2) | 32.3 (↓ 22.0) | 30.0 (↓ 21.0) |
| **EP** | ✗ | C1,C2,C8 | 66.6 | 75.8 | 63.4 | 65.3 | 67.8 |
| | ✓ | **+C7,C9** | 26.8 (↓ 39.8) | 27.1 (↓ 48.7) | 25.5 (↓ 37.9) | 27.6 (↓ 37.7) | 26.8 (↓ 41.0) |
| **FR** | ✗ | C1,C2,C3,C6,C8 | 70.9 | 89.7 | 85.6 | 85.7 | 83.0 |
| | ✓ | **+C7,C9,C10** | 41.0 (↓ 29.9) | 59.1 (↓ 30.6) | 37.0 (↓ 48.6) | 34.2 (↓ 51.5) | 42.8 (↓ 42.9) |

high scores in the smaller models, reaching 0.46. As the parameters increase, the scores decrease, but overall, the performance remains normal.

### 4.3 CAPABILITY ANALYSIS

**Atomic Capability *vs.* Compositional Capabilities.** Firstly, the Tab. 4 shows that on the 5 types of questions with more compositional abilities, the performance of the four models all drops significantly. For the 3 questions with integrating C7 (Multi-view Fusion) capability, the models' accuracy in SA (Size Assessment), OO (Object-Object Spatial Relationship), and OS (Object-Scene Spatial Relationship) tasks decreases by 19.1%, 7.6%, and 21.0%, respectively. Despite the drop, the accuracy of SA and OO still remains at a decent level. Secondly, in the EP task with both C7 and C9 (Reverse Reasoning) abilities integrated, the models' overall accuracy plummets from 67.8% without integration to 26.8%, a huge drop of 41.0%. Similarly, for the FR task, which with the most compositional capacities with an additional C10 (Situated Observation), the four models' average score crashes from 83.0% to 42.8%, an even larger drop of 42.9%.

These results partially reveal the relationship between accuracy and capability. As more abilities are incorporated, model performance declines to varying degrees, with some decreases even exceeding 50.0%. This indicates that current models have a limited grasp of integrated spatial intelligence, thereby highlighting the necessity of SpaCE-10.

**Spatial Capability Breakdown.** To better understand the strengths and weaknesses of the current MLLMs across various atomic capabilities, we constructed a model accuracy and atomic capability score matrix (Fig. 4) to associate QA accuracy with capabilities. Notably, this matrix is an unweighted average, so it diagnoses the model's capability independent of task weights in calculating accuracy. About the calculation details, the QA and capability mapping is shown in Tab. 5 and the calculation method is shown in Sec. E. Moreover, because C1 (Object Recognition), C7 (Multi-view Fusion), and C8 (Forward Thinking) appear in all eight QA types (100% coverage), their scores are the model's eight QA-type average accuracies. Our core findings are as follows.

(1) Strengths and Weaknesses: C4 (Size Comparison) is consistently the strongest atomic capability across almost all models, whereas C5 (Counting) is uniformly the weakest. This pattern indicates that continuous magnitude judgments are comparatively well handled, while discrete numeros-

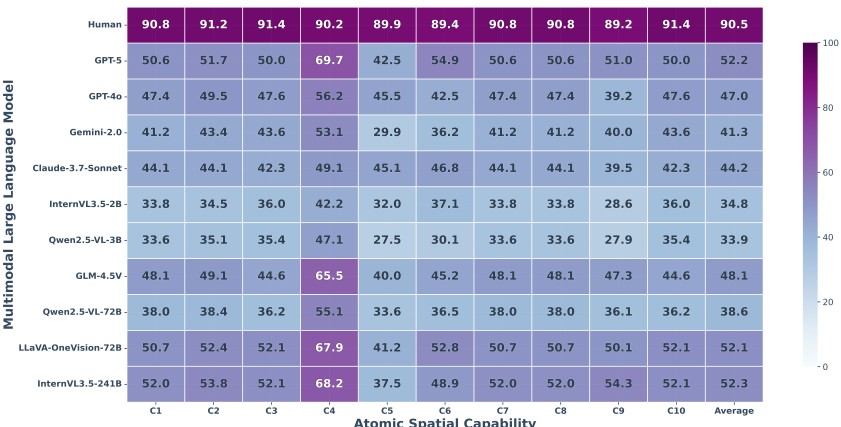

Figure 4: **Results of representative MLLMs on 10 atomic capabilities of SpaCE-10.** Each value reflects the model's average accuracy (%) across all question types involving the respective spatial capability (C1-C10), as defined in the benchmark's task-to-capability mapping.

ity under occlusion/clutter and across views remains a persistent perceptual challenge. Notably, while InternVL3.5-241B achieves the highest overall capability (52.3%) and accuracy (55.0%) average among all models, its performance on C5 (37.5%) is substantially lower than that of GPT-4o (45.5%), whose overall accuracy is 49.0%. This contrast potentially highlights that closed-source models may possess better generalization in tougher capabilities.

(2) Similar Accuracy $\neq$ Similar Capability: Interestingly, comparing GLM-4.5V, LLaVA-OneVision-72B, and InternVL3.5-241B, we observe that their task-weighted overall accuracies are 51.6%, 52.6%, and 55.0%, respectively. Yet GLM's capability average is only 48.1%, below LLaVA's 52.1%. Conversely, InternVL3.5-241B's overall accuracy exceeds LLaVA by 2.4% while its capability average differs by only 0.3%. This comparison highlights a crucial fact: high accuracy does not necessarily indicate high capability. Accuracy in Tab. 2 emphasizes 'task completion rate', whereas the Capability matrix in Fig. 4 is another dimension that reveals how MLLMs master diverse spatial capabilities. Relying solely on overall accuracy can therefore mislead about a model's generalizability. The takeaway is that future efforts to boost MLLM spatial intelligence must look beyond accuracy and shore up capabilities, such as C3 (Spatial Relationship), C10 (Situated Observation), and C5 (Counting), to achieve truly robust spatial reasoning. This divergence also validates the necessity of our capability-based analysis.

(3) Scaling improves, but does not break bottlenecks: Scaling model parameters under our evaluation framework indeed yields significant overall performance gains. For example, InternVL3.5 improves its capability average from 34.8% at 2B parameters to 52.3% at 241B parameters, showing that larger models generally master a broader set of spatial abilities. However, this scaling trend is not uniform across all capabilities. In particular, for C5 (Counting), even the largest InternVL3.5-241B achieves only 37.5%, far below its strength in other dimensions and still substantially weaker than human performance (89.9%). This indicates that parameter scaling alone struggles to deliver qualitative improvements in discrete numerical reasoning within spatial contexts. By contrast, models like GPT-4o, GLM-4.5V, and LLaVA-OneVision-72B, achieve superior scores in C5. This suggests that architectural design, training strategy, and data diversity may play a more decisive role than raw scaling for certain atomic spatial capabilities.

**Spatial Capability Improvement** Taken together, our findings point to two potential paths for improving spatial intelligence in MLLMs: (1) **Counting-focused supervision:** curate training sets that stress occlusion, crowding, and multi-view consistency to build reliable discrete counting ability in realistic scenes; (2) **Capability-aware training beyond scale:** while increasing model size improves overall accuracy and the capability average, gains on the view-conditioned relation chain of C3 (Spatial Relationship) and C10 (Situated Observation) remain limited. Thus, future work could curate training data from a capability perspective and adopt training strategies (e.g. post-training) based on different capability compositions.

## 5 CONCLUSION

In this paper, we propose SpaCE-10, a comprehensive benchmark for evaluating compositional spatial intelligence in Multimodal Large Language Models (MLLMs). SpaCE-10 covers evaluations of MLLMs from 10 atomic spatial capabilities to 8 compositional capabilities. In SpaCE-10, we collect images and point clouds from 800 scenes and design a hierarchical annotation pipeline to produce over $5k$ high-quality question-answer pairs, covering various evaluation settings of MLLMs. Through extensive evaluation on nearly 50 MLLMs, we reveal critical limitations of current MLLMs and draw several findings that are beneficial to future work in the community. We believe these studies will provide an invaluable hint for future research toward human-level machine intelligence.

## ACKNOWLEDGMENT

This work was supported by the National Natural Science Foundation of China (62506229), Natural Science Foundation of Shanghai under 25ZR1402268, Shanghai QiYuan Innovation Foundation, and Scientific Research Innovation Capability Support Project for Young Faculty (U40) of the Ministry of Education of China (SRICSPYF-ZY2025019).

## ETHICS STATEMENT

All visual data in SpaCE-10 are sourced from publicly available datasets, ensuring compliance with original data licenses and no privacy violations. Human experts involved in annotation and evaluation were informed of the task purpose, and their contributions were voluntary. We checked for potential biases across scenes and QA types to achieve balanced evaluation of specific MLLM capabilities. The benchmark and evaluation code will be released with clear usage guidelines to prevent misuse for misleading spatial reasoning applications.

## REPRODUCIBILITY STATEMENT

We provide details of the data curation and evaluation in the main text and appendix to ensure reproducibility. Our source code and data are publicly released.

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

## USE OF LARGE LANGUAGE MODELS (LLMS)

During the writing of this paper, we utilized LLM solely for language editing to improve clarity and readability. We critically reviewed and revised all AI-generated suggestions to ensure the final text accurately reflects our original intent. All intellectual contributions, including the research design, methodology, analysis, and conclusions, are our exclusive work, and we take full responsibility for the academic integrity of this publication.

## A  OVERVIEW OF THE APPENDIX

This appendix provides an introduction of QA definition and examples, atomic capability definition, calculation of correlation between character length and human accuracy, calculation of capability score matrix, experiments on data generation pipeline, case study, annotate interface, ethic statement, and reproducibility statement. It is organized as follows:

In Sec. B, we detail the definition and example of each QA type. The examples of Basic QA and Compositional QA are also illustrated in this section. In Sec. C, we introduce the definition of each atomic spatial capability and our design motivation. In Sec. D, we describe the detailed process of QA generation in Sec. D.1and demonstrate the examples of Basic QA and Compositional QA in Sec. D.2. The multiple-answer examples are also attached in Sec. D.3. In Sec. E, we demonstrate the calculation method of the capability score with a simple example. In Sec. F, we make ablation study on the effectiveness of the data generation pipeline. In Sec. G, we show the unweighted overall performance of Tab.2 in the main paper. In Sec. H, we show the case study of Basic QA quality in Sec. H.1, our curated annotation interface in Sec. H.2, and more QA cases in Sec. H.3. In Sec. I, we discuss more details in this paper. In Sec. I.6, we announce the usage of LLMs.

Table 5: **Mapping of QA types and Spatial Capabilities.** Each type of QA is the integration of multiple capabilities.

| Tasks | c1 | c2 | c3 | c4 | c5 | c6 | c7 | c8 | c9 | c10 |
|---|---|---|---|---|---|---|---|---|---|---|
| Entity Quantification (EQ) | ✓ | ✓ | - | - | ✓ | - | ✓ | ✓ | - | - |
| Scene Quantification (SQ) | ✓ | - | - | - | ✓ | ✓ | ✓ | ✓ | - | - |
| Size Assessment (SA) | ✓ | ✓ | - | ✓ | - | - | ✓ | ✓ | - | - |
| OO-Spatial Relationship (OO) | ✓ | ✓ | ✓ | - | - | - | ✓ | ✓ | - | ✓ |
| OS-Spatial Relationship (OS) | ✓ | ✓ | ✓ | - | - | - | ✓ | ✓ | - | ✓ |
| Entity Presence (EP) | ✓ | ✓ | - | - | - | - | ✓ | ✓ | ✓ | - |
| Functional Reasoning (FR) | ✓ | ✓ | ✓ | - | - | ✓ | ✓ | ✓ | ✓ | ✓ |
| Spatial Planning (SP) | ✓ | ✓ | ✓ | - | - | - | ✓ | ✓ | ✓ | ✓ |

## B QA DEFINITION

In this section, we introduce the definition of each QA type and show the QA examples in Fig. 5. Notably, each QA type is the integration of multiple capabilities, and we also show the mapping between QA and capabilities in this figure and Tab. 5. The following are definitions:

- Entity Quantification (EQ): Counting the number of objects in a scene.
- Scene Quantification (SQ): Counting the number of regions within a scene.
- Size Assessment (SA): Comparing the size relationships between different objects.
- Object-Object Spatial Relationship (OO): Understanding the relative spatial relationship between two objects.
- Object-Scene Spatial Relationship (OS): Understanding the relative spatial relationship between an object and the overall scene.
- Functional Reasoning (FR): Reasoning objects that match or do not match certain functions based on relative spatial relationships.
- Spatial Planning (SP): Global spatial navigation and path planning.

## C CAPABILITY DEFINITION

In this section, we illustrate the definition of each capability:
C1-Object Recognition: Identify what the object is.
C2-Spatial Location: Localizing an object in space.
C3-Spatial Relationship: Understanding relative spatial position relationship.
C4-Size Comparison: Comparing the size relationship of objects.
C5-Counting: Count the number of objects and scenes.
C6-Function Knowledge: Understanding the function of objects.
C7-Multi-view Fusion: Understanding spatial information from multiple views
C8-Forward Thinking: Understand the forward instructions and complete tasks in the given space.
C9-Reverse Reasoning: Understand the reverse instructions and complete tasks in the given space.
C10-Situated Observation: Imagine standing in a designated position in space and observing and understanding the scene.

In designing our capabilities, we draw on established research in spatial intelligence. Initially, a beginning reference for our design is Sparkle (Tang et al., 2024), which frames 2D spatial intelligence around three abilities—Direction, Distance, and Location. In SpaCE-10, the direction and distance are split into C3 (Spatial Relationship). Since we believe that the concept of 'what and where' is one of the most fundamental abilities for real spatial intelligence, location is separated and mapped to the C2 (Spatial Location). Next, early studies (Kamath et al., 2023; Bagherinezhad et al., 2016) mainly emphasize C4 (Size Comparison) and simple positional cues in clean images. Then, as evaluation expands from single-scene 3D point-cloud QA (Azuma et al., 2022; Ma et al., 2022;

**Entity Presence:**

**Q:** Which of the following options does not contain any objects in the given scene?

**A.** lush green plant, brown TV, blue bed, gray cabinet

**B.** brown lamp, blue carpet, gray TV, gray window

**C.** brown door, gray TV, gray window, blue cabinet

**D.** lush green plant, black TV, blue bed, black window

**Capability:** C1, C2, C7, C8, C9

**Spatial Planning:**

**Q:** Which of the following descriptions are incorrect for keeping the room clean and tidy?

**A.** From the entrance first move to the room center, turn left to the bed, then tidy up the quilt.

**B.** From the entrance first move to the room center, turn right to the bed, then tidy up the quilt.

**C.** From the entrance first move to the room center, turn left to the cabinet, then clean up the cabinet.

**D.** From the entrance first move to cabinet, turn left to the smaller room, then close the window.

**Capability:** C1, C2, C3, C7, C8, C9, C10

**Entity Quantification:**

**Q:** How many beds are in the given scene?

**A.** 1 **B.** 2 **C.** 3 **D.** 4

**Capability:** C1, C2, C5, C7, C8

**Scene Quantification:**

**Q:** The area indicated by objects with specific functions is called a functional zone (for example, a desk in the workspace, chairs and tables in the dining area). In a given scene, how many spatially different functional zones are there? **A.** 1 **B.** 2 **C.** 3 **D.** 4

**Capability:** C1, C5, C6, C7, C8

**Size Assessment:**

**Q:** Which description is correct?

**A.** The door is lower than the bed.

**B.** The door is wider than the TV.

**C.** The lamp is taller than the bed.

**D.** The plant is lower than the bed.

**Capability:** C1, C2, C4, C7, C8

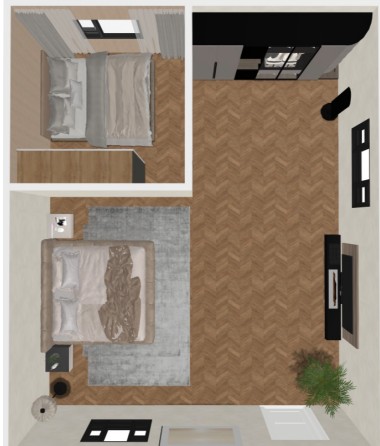

**OO Spatial Relationship:**

**Q:** When facing the front side of the objects, which descriptions is correct?

**A.** The plant is on the left of the TV.

**B.** The plant is on the right of the TV.

**C.** The lamp is on the right of the bed.

**D.** The mural is on the right of the door.

**Capability:** C1, C2, C3, C7, C8, C10

**OS Spatial Relationship:**

**Q:** When viewing from the entrance, which descriptions are correct?

**A.** Bed is on the center of the room.

**B.** TV is on the left side of the room.

**C.** The cabinet is the farthest object that can be seen.

**D.** Bed is on the right side of the room.

**Capability:** C1, C2, C3, C7, C8, C10

**Function Reasoning:**

**Q:** Which of the following description is correct?

**A.** Among the objects adjacent to the bed, there is an object can be used to store items, but lacks two objects that can be used to clean the floor and to provide ventilation.

**B.** Among the objects adjacent to the cabinet, there is an object can be used to check appearance, but lacks two objects that can be used to provide ventilation and to rest.

**C.** Among the objects adjacent to the TV, there is an object can be used to check appearance, but lacks two objects that can be used to provide ventilation and to decorate the room.

**D.** Among the objects adjacent to the TV, there is an object can be used to decorate the room, but lacks two objects that can be used to clean the floor and to provide ventilation.

**Capability:** C1, C2, C3, C6, C7, C8, C9, C10

Figure 5: **Examples of all types of QA.** The blue examples represent the perception QA, and the purple ones denote the reasoning QA. The green circles are the correct answer.

Ye et al., 2022; Zhao et al., 2022; Yan et al., 2023; Hong et al., 2023; Yang et al., 2025a; Lyu et al., 2024; Huang et al., 2025a;b) to multi-view 2D imagery (Ma et al., 2024; 2025; Linghu et al., 2024; Yang et al., 2024; 2025b; Jia et al., 2025; Cheng et al., 2024), the focus shifts from single-view C3 (spatial localization) and C5 (Counting) within one scene to cross-view consistency captured by C7 (Multi-view Fusion). In parallel, SQA (Ma et al., 2022) introduces viewpoint conditioning, aligning with C10 (Situated Observation). Nowadays, with the rise of Embodied Intelligence and Reasoning MLLMs, based on these previous excellent works, we further propose to incorporate C6 (Function Knowledge), C8 (Forward Thinking), and C9 (Reverse Reasoning) into this atomic capability pool for manipulating basic knowledge and reasoning ability examination.

# D  DATA GENERATION DETAILS

## D.1  GENERATION FOR EQ AND SP

For EQ, we first extract the number of each object from the scan datasets' semantic labels, and then leverage a predefined template to generate QA. For the SP, we employed human experts to manually design 80 QA pairs. Each question presents a complete task flow, including the navigation path, goal, goal characteristics, and actions to be performed, with potential errors in any step. Based on the prompt, the model must select either a fully correct task flow or one containing incorrect steps.

## D.2 Basic QA and Compositional QA

In the main paper, we mention that there are 5 QA types (OO, OS, SA, FR, SP) that will be applied with cross-capability integration strategy. Thus, in this section, we demonstrate examples of these two QAs.

(1) For OO (Object-Object Spatial Relationship), OS (Object-Scene Spatial Relationship), SA (Size Assessment):

**Basic-SA-Question:** When facing the front side of the objects, which description is correct?

**Basic-SA-Options:**

A. This red table is taller than the brown cabinet, but narrower than the brown cabinet.

B. This red table is shorter and narrower than the brown cabinet.

C. This red table is shorter, but wider than the brown cabinet.

D. This red table is taller than the brown cabinet and wider than brown cabinet.

**Basic-SA-Answer:** B

In the Basic QA format, each question involves only two objects, and the scenario is restricted to a single viewpoint, i.e., the front of the objects. For each scene, we generate multiple Basic QA questions, and then aggregate these options to form Compositional QA questions.

**Compositional-SA-Question:** When facing the front side of the objects, which description is correct?

**Compositional-SA-Options:**

A. This red table is taller than the brown cabinet, but narrower than brown cabinet.

B. The blue sofa is wider than the white door.

C. The green plant is taller and wider than the wooden bookshelf.

D. This red table is taller than the brown cabinet and wider than brown cabinet.

**Compositional-SA-Answer:** B

In Compositional QA, we combine multiple Basic QA questions, so each question may now refer to different objects located at various positions across the scene. This forces the model to integrate information from multiple perspectives, requiring C7: Multi-view Fusion.

(2) For EP (Entity Presence), we add additional C7 and C9:

**Basic-EP-Question 1:** Does a red chair exist in the given scene?

**Basic-EP-Options 1:**

A. Yes B. No

**Basic-EP-Answer 1:** A

**Basic-EP-Question 2:** Does a gray sofa exist in the given scene?

**Basic-EP-Options 2:**

A. Yes B. No

**Basic-EP-Answer 2:** B

We generate multiple Basic-EP questions and then aggregate them into Compositional-EP questions.

**Compositional-EP-Question:** Which of the following options does not contain any objects in the given scene?

**Compositional-EP-Options:**

A. Red chair, blue sofa, green plant, red table

B. Gray sofa, white lamp, orange carpet, pink cushion

C. Black coffee table, grey armchair, purple curtain, brown bookshelf

D. Beige ottoman, teal vase, silver lamp, golden picture frame

**Compositional-EP-Answer:** B

In Compositional-EP, each option contains multiple objects. Similar to Basic QA, where objects in the scene may appear from different angles or positions, C7: Multi-view Fusion is required. Additionally, the task shifts from a direct "does this object exist?" question to a more complex "which object is missing from the scene?" requiring C9: Reversed Reasoning.

(3) For FR (Functional Reasoning), we add additional C7, C9, and C10:

**Basic-FR-Question:** Which option correctly describes the function of an object near the bed?

**Basic-FR-Options:**

A. Cabinet, store items

B. Mop, clean the floor

C. Window, provide ventilation

D. Mop, provide light

**Basic-FR-Answer:** A

In Basic-FR, the correct answer must satisfy both functional and spatial positioning requirements. We generate multiple Basic-FR questions, which are then combined into Compositional-FR questions.

**Compositional-FR-Question:** Which of the following descriptions is correct?

**Compositional-FR-Options:**

A. Among the objects adjacent to the bed, there is an object that can be used to store items, but lacks two objects that can be used to clean the floor and provide ventilation.

B. Among the objects adjacent to the cabinet, there is an object that can be used to check our appearance, but lacks two objects that can be used to provide ventilation and rest.

C. Among the objects adjacent to the TV, there is an object that can be used to check our appearance, but lacks two objects that can be used to provide ventilation and decorate the room.

D. Among the objects adjacent to the TV, there is an object that can be used to decorate the room, but lacks two objects that can be used to clean the floor and provide ventilation.

**Compositional-FR-Answer:** A

In Compositional-FR, multiple central objects are combined, broadening the question to encompass the entire scene. The model must analyze the scene from various perspectives, necessitating C7: Multi-view Fusion to integrate spatial relationships and C10: Situated Observation to understand the contextual functionality of objects. Furthermore, the task involves evaluating the relevance of functional descriptions, which requires C9: Reverse Reasoning to assess the appropriateness of the functions in relation to the scene.

### D.3    MULTI-ANSWER EXAMPLE

Each multiple-choice question is set as a five-option, two-correct answer question, where the evaluated model must select two correct answers to be considered correct. The accuracy for each type of QA is calculated in the same way as single-choice questions. Compared to single-choice questions, the question and option format for multiple-choice questions remains exactly the same. Below is an example of the EP:

**Single-Choice EP:**

Question: Which of the following options does not contain any objects in the given scene?

Options:

A. Red chair, blue sofa, green plant, red table

B. Gray sofa, white lamp, orange carpet, pink cushion

C. Black coffee table, grey armchair, purple curtain, brown bookshelf

D. Beige ottoman, teal vase, silver lamp, golden picture frame

Answer: B

Double-Choice EP:

Question: Which of the following options does not contain any objects in the given scene?

Options:

A. Red chair, blue sofa, green plant, red table

B. Gray sofa, white lamp, orange carpet, pink cushion

C. Black coffee table, grey armchair, purple curtain, brown bookshelf

D. Beige ottoman, teal vase, silver lamp, golden picture frame

E. Gray sofa, grey armchair, purple curtain, brown bookshelf

Answer: B, E

## E  CAPABILITY SCORE CALCULAION

**Capability-score computation (for Fig. 4).**  The values in Fig. 4 are computed from the per-question scores in Tab. 5 via the task→capability mapping (C1-C10).

Let $\mathcal{Q}_i$ denote the set of questions linked to capability $C_i$ and $n_i = |\mathcal{Q}_i|$. For a question $q \in \mathcal{Q}_i$, let $\mathrm{Score}(q)$ be the score assigned by the task's evaluation rule (e.g., 0/100 for single-answer exact match, or the task-specific percentage for multiple-answer). The capability score is the mean over its linked questions:

$$\mathrm{Score}(C_i) \;=\; \frac{1}{n_i} \sum_{q \in \mathcal{Q}_i} \mathrm{Score}(q). \tag{1}$$

If no question maps to a capability ($n_i = 0$), the entry is marked as N/A and excluded from any further averaging. When a question maps to multiple capabilities, its score contributes to each linked capability (no reweighting).

**Toy example.**  Consider two capabilities $C_1$ and $C_2$, and two questions $Q_1, Q_2$ with mappings $Q_1 \mapsto C_1$ and $Q_2 \mapsto \{C_1, C_2\}$. Suppose $\mathrm{Score}(Q_1) = 80$ and $\mathrm{Score}(Q_2) = 90$. Then $\mathcal{Q}_1 = \{Q_1, Q_2\}$ ($n_1 = 2$) and $\mathcal{Q}_2 = \{Q_2\}$ ($n_2 = 1$), yielding

$$\mathrm{Score}(C_1) = \tfrac{80+90}{2} = 85, \qquad \mathrm{Score}(C_2) = 90.$$

This procedure produces the capability values shown in Fig. 4.

## F  PIPELINE COMPONENT ABLATION

To systematically evaluate the contribution of each key component in the structured data generation pipeline to the accuracy of basic QA generation, we designed a stepwise ablation study. Specifically, we selected five QA task types (SA, OO, OS, EP, FP), and for each type, we randomly sampled 30 scenes, generating one question per scene. The accuracy of each generated question was manually verified by human experts.

The experimental results in Tab. F clearly demonstrate the cumulative gains of each module. Using the 2D Captioner alone as the baseline, the generated QA already showed relatively stable accuracy across most categories (an average of 62.0%), with particularly high accuracy in the EP task at 96.7%. This reflects the relatively low difficulty of generating questions for this category and that

Table 6: **Ablation study of structural data generation pipeline.** We randomly sample 30 scenes for each type of QA and generate one QA for each scene. The results demonstrate the effectiveness of each component.

| Component | SA | OO | OS | EP | FP | Average |
|---|---|---|---|---|---|---|
| 2D Captioner | 50.0 | 50.0 | 46.7 | 96.7 | 66.7 | 62.0 |
| + 3D Captioner | 80.0 | 63.3 | 63.3 | 96.7 | 80.0 | 76.7 (+14.7) |
| + Inspector | 80.0 | 66.7 | 70.0 | 96.7 | 83.3 | 79.3 (+17.3) |
| + Structural Data | 86.7 | 76.7 | 76.7 | 96.7 | 83.3 | 84.0 (+22.0) |

2D visual information sufficiently supports it. With the addition of the 3D Captioner, the overall accuracy improved significantly (an average increase of 14.7%), indicating that 3D information effectively supplements the limitations of 2D vision and enhances the model's understanding of spatial and object attributes. Further incorporating the Inspector component led to another increase in accuracy, reaching 79.3% (a 17.3% improvement over the baseline), showing that this module plays an important role in validating and refining the details of question generation. Finally, after adding structured data, the overall average accuracy reached 84.0%, a 22% improvement compared to the initial baseline, fully demonstrating the critical value of structured information in improving the quality of basic QA generation.

# G    UNWEIGHTED MAIN RESULTS

We demonstrate the unweighted single-answer performance ranking on SpaCE-10 in Tab. 7.

# H    CASE STUDY

## H.1    BASIC QA QUALITY

In this paper, we have developed a sophisticated pipeline for the automated generation of high-quality Basic QA (Basic Question Answering). In the previous section, we systematically verified the effectiveness of each component within the pipeline. This section presents two typical high-quality QA cases and delves into them with specific context-based analysis and discussion.

As shown in Fig. 6, the first case centers on the size estimation task (Basic SA). As depicted in the figure, through meticulously crafted prompts and high-quality snapshots, we steered GPT to accurately generate questions. When comparing the volume relationship between a white wooden baby crib and a wardrobe, GPT delivered an impressive performance. It not only correctly identified the height and width dimensions of each object but also precisely determined their relative differences across multiple perspectives. For instance, its response accurately pointed out that the wardrobe is taller than the crib but has a similar width, demonstrating a good grasp of the 3D geometric properties of objects.

The second case focuses on the spatial relationship understanding task (Basic OO), as shown in the Fig. 7, in the context of judging the spatial relationship between a yellow chair and an entrance. In this example, GPT not only accurately distinguished the relative direction of "left" but also correctly identified the spatial distance difference between "near" and "far". Although the question only involved one yellow chair, the distractors in the options were somewhat deceptive, which in turn enhanced the question's ability to assess the model's spatial understanding. This case also indirectly confirms the feasibility and rationality of using GPT for generating spatial intelligence QA questions. Through these two cases, we observed that powerful MLLMs, when only provided with high-quality 2D visual inputs, are already capable of understanding certain 3D spatial information, such as object size, volume relationships, and spatial orientation. This ability suggests a promising path for future exploration: unlike current 3D models that sacrifice conversational abilities to fit point cloud data, 2D MLLMs can still demonstrate strong spatial understanding potential even without explicitly incorporating 3D structural modeling.

Table 7: **Unweighted overall performance ranking on SpaCE-10.**

| Models | Rank | Perception | | | | | Reasoning | | | Overall |
|---|---|---|---|---|---|---|---|---|---|---|
| | | EQ | SQ | SA | OO | OS | EP | FR | SP | |
| Human | 1 | 91.3 | 88.5 | 90.2 | 93.4 | 95.6 | 91.1 | 90.3 | 86.3 | 90.8 |
| **3D MLLMs** | | | | | | | | | | |
| LEO-7B (Huang et al., 2023) | 46 | 15.8 | 0.0 | 16.7 | 16.5 | 25.2 | 5.5 | 5.7 | 13.3 | 12.3 |
| GPT4Scene-7B (Qi et al., 2025) | 31 | 30.9 | 37.7 | 38.0 | 38.9 | 41.6 | 29.5 | 28.0 | 32.5 | 34.6 |
| **Close Source 2D MLLMs** | | | | | | | | | | |
| GPT-5 (OpenAI, 2025) | 5 | 42.0 | 43.0 | 69.7 | 60.7 | 36.5 | 50.3 | 66.8 | 30.0 | 49.9 |
| GPT-4o (Achiam et al., 2023) | 9 | 58.3 | 32.8 | 56.2 | 58.3 | 56.2 | 41.6 | 52.2 | 23.7 | 47.4 |
| Gemini-2.0 (Team et al., 2023) | 19 | 34.3 | 25.6 | 53.1 | 42.6 | 53.8 | 42.2 | 46.7 | 31.2 | 41.2 |
| Claude-3.7-Sonnet (cla) | 14 | 46.0 | 44.3 | 49.1 | 46.0 | 49.1 | 44.3 | 49.3 | 25.0 | 44.1 |
| **Open Source 2D MLLMs** | | | | | | | | | | |
| ▼ *Scale < 7B* | | | | | | | | | | |
| InternVL2.5-1B (Chen et al., 2024) | 33 | 33.0 | 54.1 | 18.8 | 43.6 | 29.9 | 26.7 | 41.0 | 23.7 | 33.9 |
| InternVL3-1B (Zhu et al., 2025) | 25 | 30.7 | 55.7 | 27.9 | 44.6 | 31.6 | 47.8 | 41.9 | 30.0 | 38.8 |
| InternVL3.5-1B (Wang et al., 2025) | 34 | 34.8 | 41.7 | 29.4 | 42.7 | 25.9 | 21.9 | 40.2 | 33.8 | 33.8 |
| InternVL2.5-2B (Chen et al., 2024) | 41 | 32.2 | 26.8 | 27.0 | 36.6 | 28.8 | 21.7 | 48.2 | 36.2 | 32.2 |
| InternVL3-2B (Zhu et al., 2025) | 15 | 41.5 | 45.9 | 45.4 | 45.7 | 31.9 | 45.7 | 48.7 | 41.3 | 43.3 |
| InternVL3.5-2B (Wang et al., 2025) | 35 | 35.6 | 28.4 | 42.2 | 45.7 | 32.3 | 20.1 | 45.8 | 20.0 | 33.8 |
| Qwen2.5-VL-3B-Instruct | 37 | 31.7 | 23.3 | 47.1 | 51.7 | 31.6 | 25.5 | 37.0 | 21.2 | 33.6 |
| SpaceOm$^\diamond$ | 39 | 21.8 | 24.5 | 47.3 | 49.7 | 32.7 | 21.9 | 36.7 | 25.0 | 32.5 |
| SpaceQwen$^\diamond$ | 32 | 31.2 | 26.1 | 41.2 | 52.3 | 35.2 | 28.4 | 36.4 | 22.5 | 34.2 |
| SpaceThinker$^\diamond$ | 36 | 32.7 | 22.4 | 46.7 | 50.5 | 33.4 | 22.4 | 36.9 | 24.2 | 33.6 |
| VILA1.5-3B (Lin et al., 2024) | 44 | 25.0 | 9.1 | 31.7 | 34.6 | 31.6 | 35.3 | 12.9 | 13.3 | 26.7 |
| InternVL2.5-4B (Chen et al., 2024) | 28 | 34.3 | 23.4 | 50.2 | 50.8 | 16.2 | 21.7 | 56.0 | 33.7 | 35.8 |
| MiniCPM-v4-4B (Yao et al., 2025) | 26 | 38.1 | 32.7 | 41.1 | 49.0 | 36.5 | 29.3 | 50.0 | 30.0 | 38.3 |
| InternVL3.5-4B (Wang et al., 2025) | 29 | 38.9 | 12.9 | 48.7 | 50.7 | 27.9 | 33.9 | 37.0 | 35.0 | 35.6 |
| ▼ *Scale ≤ 14B* | | | | | | | | | | |
| Qwen2.5-VL-7B-Instruct (Bai et al., 2025) | 38 | 32.7 | 36.9 | 36.9 | 35.3 | 32.3 | 27.6 | 34.2 | 27.5 | 32.9 |
| LLaVA-v1.5-7B (Liu et al., 2023) | 42 | 31.2 | 31.3 | 30.5 | 35.7 | 22.9 | 10.7 | 57.4 | 23.7 | 31.5 |
| LLaVA-OneVision-7B (Li et al., 2024) | 18 | 37.4 | 33.8 | 46.4 | 57.3 | 34.5 | 43.3 | 61.6 | 21.2 | 41.9 |
| MiMo-VL-RL-8B (Xiaomi, 2025) | 30 | 23.7 | 35.0 | 46.4 | 41.3 | 34.7 | 32.2 | 32.5 | 36.1 | 35.2 |
| Cambrian-8B (Tong et al., 2024a) | 43 | 22.6 | 18.6 | 34.8 | 32.6 | 32.3 | 25.1 | 41.4 | 23.7 | 28.9 |
| VILA1.5-8B (Lin et al., 2024) | 45 | 25.7 | 8.2 | 27.5 | 32.7 | 17.2 | 12.4 | 26.7 | 23.7 | 21.8 |
| InternVL2.5-8B (Chen et al., 2024) | 20 | 33.2 | 36.0 | 50.0 | 55.0 | 33.6 | 27.1 | 59.1 | 32.5 | 40.8 |
| InternVL3-8B (Zhu et al., 2025) | 24 | 36.6 | 29.5 | 42.9 | 51.7 | 34.5 | 26.6 | 60.6 | 37.5 | 40.0 |
| InternVL3.5-8B (Wang et al., 2025) | 23 | 37.1 | 28.5 | 61.7 | 49.8 | 35.4 | 17.6 | 54.8 | 36.3 | 40.1 |
| Gemma3-12B (Team et al., 2025a) | 21 | 41.8 | 41.2 | 55.1 | 46.5 | 35.6 | 25.0 | 53.2 | 27.5 | 40.7 |
| InternVL3-14B (Zhu et al., 2025) | 12 | 39.7 | 28.7 | 54.4 | 58.1 | 38.1 | 51.3 | 56.6 | 35.0 | 45.2 |
| InternVL3.5-14B (Wang et al., 2025) | 11 | 41.0 | 47.6 | 65.3 | 52.1 | 34.5 | 45.4 | 54.3 | 30.0 | 46.3 |
| ▼ *14B<Scale< 72B* | | | | | | | | | | |
| InternVL3.5-20B-A4B (Wang et al., 2025) | 7 | 37.4 | 43.1 | 64.1 | 58.7 | 41.4 | 54.1 | 57.6 | 28.8 | 48.2 |
| InternVL2.5-26B (Chen et al., 2024) | 17 | 34.3 | 29.3 | 62.6 | 55.4 | 33.0 | 29.2 | 61.8 | 33.7 | 42.4 |
| Gemma3-27B (Team et al., 2025a) | 22 | 39.4 | 21.7 | 63.5 | 48.5 | 37.8 | 33.2 | 51.5 | 30.0 | 40.7 |
| Qwen2.5-VL-32B-Instruct (Bai et al., 2025) | 40 | 19.9 | 26.5 | 48.9 | 36.8 | 32.3 | 31.1 | 30.1 | 32.5 | 32.3 |
| InternVL2.5-38B (Chen et al., 2024) | 13 | 38.1 | 36.1 | 64.4 | 54.3 | 36.8 | 27.4 | 63.0 | 37.5 | 44.7 |
| InternVL3-38B (Zhu et al., 2025) | 6 | 36.3 | 41.6 | 69.5 | 60.1 | 36.3 | 58.6 | 60.8 | 35.0 | 49.8 |
| InternVL3.5-38B (Wang et al., 2025) | 16 | 42.3 | 28.4 | 62.8 | 59.1 | 37.6 | 25.4 | 59.8 | 28.8 | 43.0 |
| ▼ *Scale ≥ 72B* | | | | | | | | | | |
| GLM-4.5V (Team et al., 2025b) | 8 | 38.9 | 41.1 | 65.5 | 61.1 | 36.7 | 61.2 | 49.3 | 31.3 | 48.1 |
| LLaVA-OneVision-72B (Li et al., 2024) | 4 | 44.1 | 38.3 | 67.9 | 64.5 | 40.3 | 46.7 | 67.3 | 36.2 | 50.7 |
| Qwen2.5-VL-72B-Instruct (Bai et al., 2025) | 27 | 32.4 | 34.9 | 55.7 | 40.9 | 32.1 | 36.5 | 38.0 | 33.7 | 38.0 |
| InternVL2.5-78B (Chen et al., 2024) | 10 | 27.8 | 45.0 | 62.4 | 64.4 | 40.3 | 23.7 | 67.3 | 40.0 | 46.4 |
| InternVL3-78B (Zhu et al., 2025) | 3 | 36.8 | 48.2 | 65.3 | 61.6 | 43.8 | 44.4 | 64.3 | 46.3 | 51.3 |
| InternVL3.5-241B-A28B (Wang et al., 2025) | 2 | 35.8 | 39.1 | 68.2 | 63.5 | 46.2 | 64.2 | 58.6 | 40.0 | 52.0 |

$^\diamond$ Models proposed by RemyxAI (SpaceVLMs series, https://huggingface.co/remyxai/SpaceQwen2.5-VL-3B-Instruct).

## H.2 ANNOTATION INTERFACE

To facilitate efficient annotation, we developed a custom annotation tool, with the interface shown in Fig. 8. During the annotation process, human experts are restricted to viewing only 3D snapshots to judge the correctness of each QA pair; they are not allowed to view the 2D images. This design offers two main advantages: (1) It ensures a high annotation speed. By examining only a small number of 3D snapshots, annotators can quickly grasp the overall layout of the scene while significantly reducing their visual workload. (2) Since 2D images contain overly fine-grained details, many of which may not be present in the 3D scene, using only 3D information to filter out incorrect questions helps ensure that the resulting QA pairs are suitable for evaluation across both 2D and 3D models. Additionally, during evaluation, we tag erroneous questions to ensure none are overlooked. This end-to-end process not only prioritizes annotation efficiency but also reflects our rigorous commitment to data quality control.

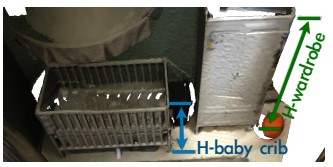
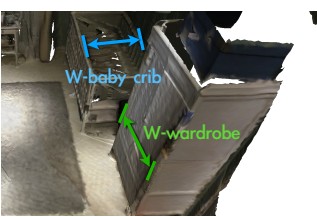

**Question:** Which of the following statements accurately compares the relative sizes of a white wooden baby crib and a wardrobe?

A. The wardrobe is taller than the white wooden baby crib, 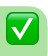 but they are similar in width.

B. The white wooden baby crib is both taller and wider than the wardrobe.

C. The wardrobe and the white wooden baby crib are the same height, but differ slightly in width.

D. The white wooden baby crib is significantly larger than the wardrobe in both height and width.

Figure 6: **Showcase of Basic SA QA.** 'H' and 'W' mean the height and width of objects, respectively. In this case, GPT-4o shows the precise understanding of both size and geometry.

**Question:** Which of the following options best describes the spatial relationship between the yellow chair and the entrance in the provided 3D scene image?

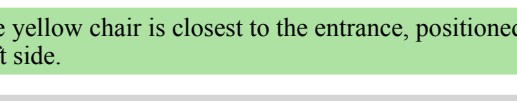

A. The yellow chair is closest to the entrance, positioned on the left side.

B. The yellow chair is farthest from the entrance, positioned on the right side.

C. The yellow chair is arranged at the back of the room in front of the whiteboard.

D. The yellow chair is placed diagonally opposite the entrance, near the black chair.

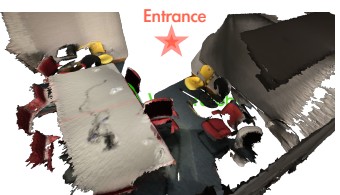

Figure 7: **Showcase of Basic OO.** This case reflects the capability in spatial relationships and QA design. Notably, all spatial relationship is based on the observation at the entrance.

### H.3 VISUALIZATION OF QA

We demonstrate more QA cases in this section. Specifically, figures of EP are shown in Fig. 9 and 10; figures of OO are shown in Fig. 11 and 12; figures of OS are shown in Fig. 13 and 14; figures of SA are shown in Fig. 15 and 16; figures of FR are shown in Fig. 17 and 18; figures of SP are shown in Fig. 19 and 20; figures of EQ and SQ are shown in Fig. 21 and 22.

## I DISCUSSION

### I.1 ERROR PATTERN ANALYSIS BASED ON VISUALIZATIONS

(1) **Analysis in EP (Entity Presence):** In the Entity Presence (EP) task, the dominant failure mode clearly aligns with C7 (Multi-view Fusion). This task requires the model to accurately determine which objects are absent from the given set of images. As shown in Fig. 9, GPT-5 selects option B, but option B explicitly mentions "a set of black-and-white bookshelves with a clean and organized structure," which is indeed present in the scene.

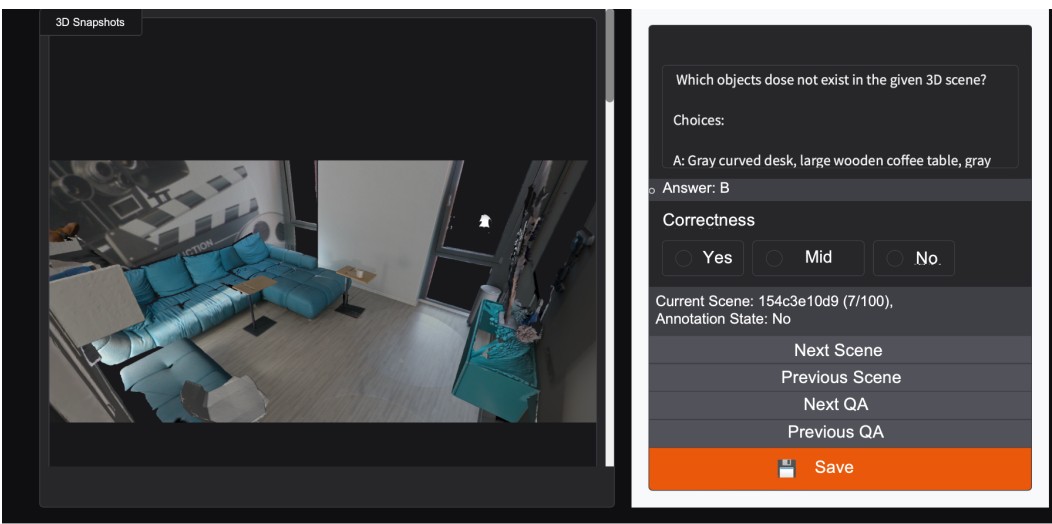

Figure 8: **Interface of annotation tools.**

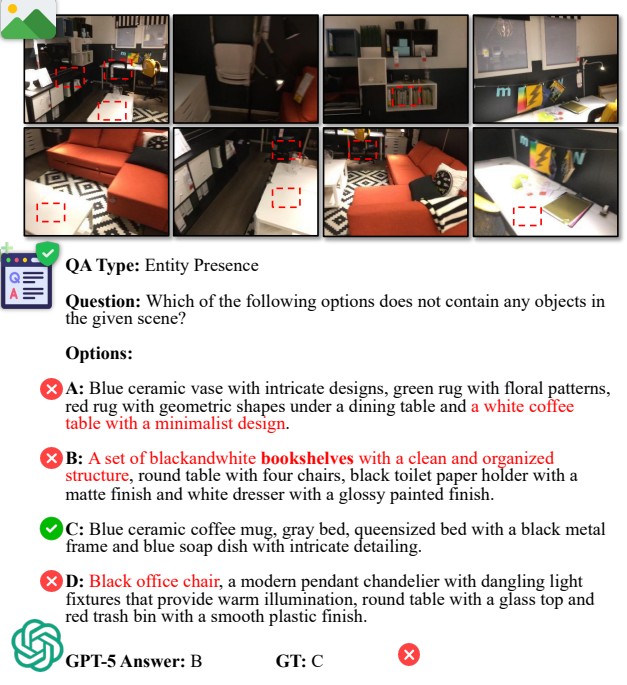

**QA Type:** Entity Presence

**Question:** Which of the following options does not contain any objects in the given scene?

**Options:**

**A:** Blue ceramic vase with intricate designs, green rug with floral patterns, red rug with geometric shapes under a dining table and a white coffee table with a minimalist design.

**B:** A set of blackandwhite **bookshelves** with a clean and organized structure, round table with four chairs, black toilet paper holder with a matte finish and white dresser with a glossy painted finish.

**C:** Blue ceramic coffee mug, gray bed, queensized bed with a black metal frame and blue soap dish with intricate detailing.

**D:** Black office chair, a modern pendant chandelier with dangling light fixtures that provide warm illumination, round table with a glass top and red trash bin with a smooth plastic finish.

**GPT-5 Answer:** B        **GT:** C

Figure 9: Visualization of Entity Presence

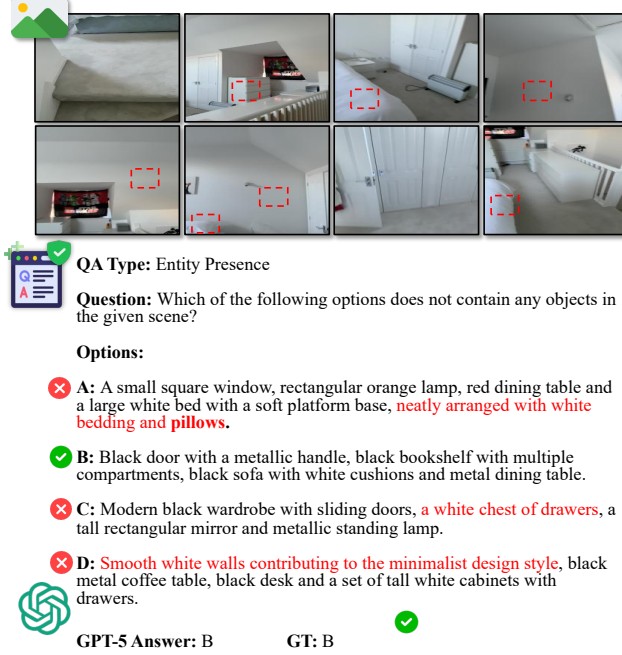

**QA Type:** Entity Presence

**Question:** Which of the following options does not contain any objects in the given scene?

**Options:**

❌ **A:** A small square window, rectangular orange lamp, red dining table and a large white bed with a soft platform base, neatly arranged with white bedding and **pillows**.

✅ **B:** Black door with a metallic handle, black bookshelf with multiple compartments, black sofa with white cushions and metal dining table.

❌ **C:** Modern black wardrobe with sliding doors, a white chest of drawers, a tall rectangular mirror and metallic standing lamp.

❌ **D:** Smooth white walls contributing to the minimalist design style, black metal coffee table, black desk and a set of tall white cabinets with drawers.

**GPT-5 Answer:** B          **GT:** B  ✅

Figure 10: Visualization of Entity Presence

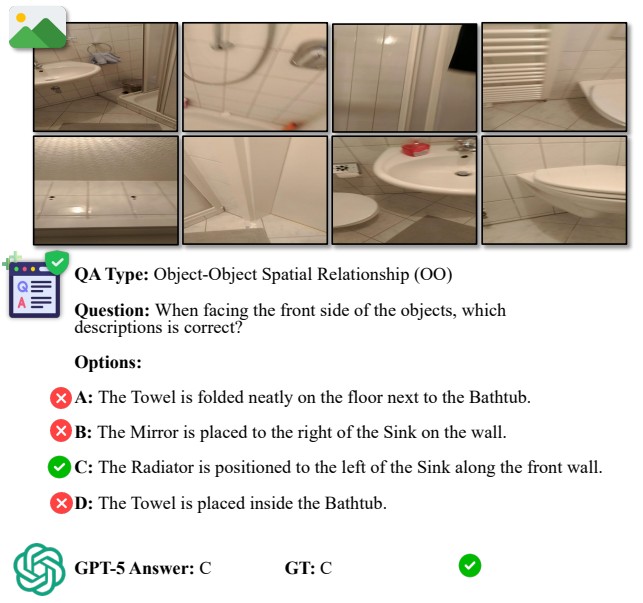

**QA Type:** Object-Object Spatial Relationship (OO)

**Question:** When facing the front side of the objects, which descriptions is correct?

**Options:**

❌ **A:** The Towel is folded neatly on the floor next to the Bathtub.

❌ **B:** The Mirror is placed to the right of the Sink on the wall.

✅ **C:** The Radiator is positioned to the left of the Sink along the front wall.

❌ **D:** The Towel is placed inside the Bathtub.

**GPT-5 Answer:** C          **GT:** C          ✅

For A and D, towel is on the wall; For B, mirror is above sink.

Figure 11: Visualization of Object-Object Spatial Relationship

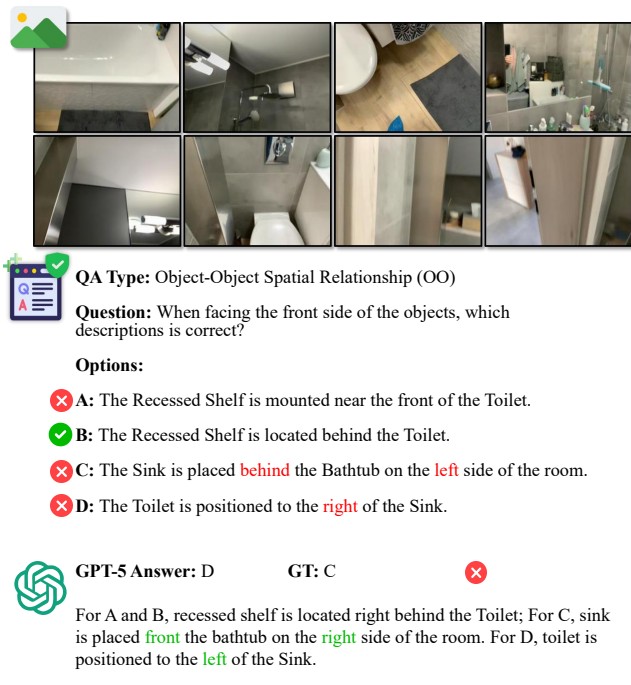

**QA Type:** Object-Object Spatial Relationship (OO)

**Question:** When facing the front side of the objects, which descriptions is correct?

**Options:**

❌ **A:** The Recessed Shelf is mounted near the front of the Toilet.

✅ **B:** The Recessed Shelf is located behind the Toilet.

❌ **C:** The Sink is placed behind the Bathtub on the left side of the room.

❌ **D:** The Toilet is positioned to the right of the Sink.

**GPT-5 Answer:** D          **GT:** C          ❌

For A and B, recessed shelf is located right behind the Toilet; For C, sink is placed front the bathtub on the right side of the room. For D, toilet is positioned to the left of the Sink.

Figure 12: Visualization of Object-Object Spatial Relationship

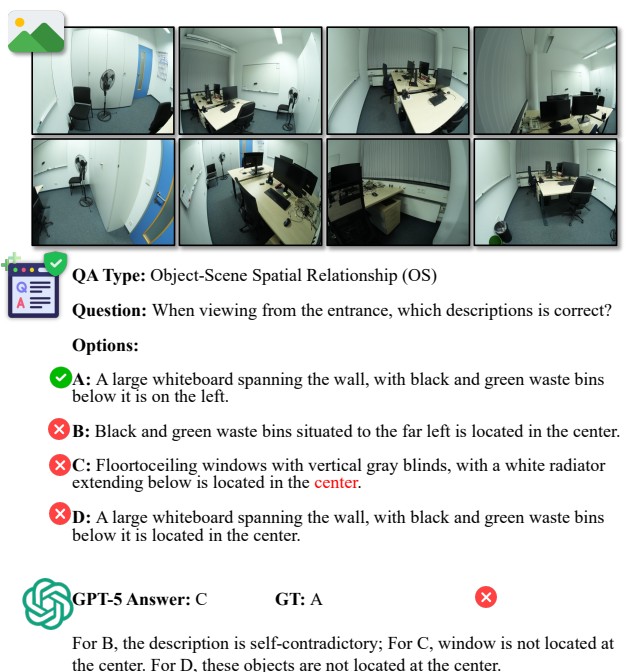

**QA Type:** Object-Scene Spatial Relationship (OS)

**Question:** When viewing from the entrance, which descriptions is correct?

**Options:**

✅ **A:** A large whiteboard spanning the wall, with black and green waste bins below it is on the left.

❌ **B:** Black and green waste bins situated to the far left is located in the center.

❌ **C:** Floortoceiling windows with vertical gray blinds, with a white radiator extending below is located in the center.

❌ **D:** A large whiteboard spanning the wall, with black and green waste bins below it is located in the center.

**GPT-5 Answer:** C          **GT:** A          ❌

For B, the description is self-contradictory; For C, window is not located at the center. For D, these objects are not located at the center.

Figure 13: Visualization of Object-Scene Spatial Relationship

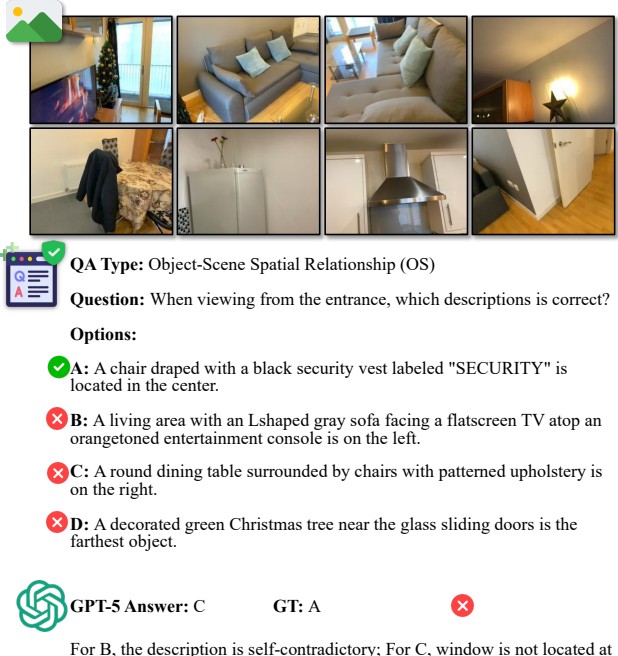

**QA Type:** Object-Scene Spatial Relationship (OS)

**Question:** When viewing from the entrance, which descriptions is correct?

**Options:**

**A:** A chair draped with a black security vest labeled "SECURITY" is located in the center.

**B:** A living area with an Lshaped gray sofa facing a flatscreen TV atop an orangetoned entertainment console is on the left.

**C:** A round dining table surrounded by chairs with patterned upholstery is on the right.

**D:** A decorated green Christmas tree near the glass sliding doors is the farthest object.

**GPT-5 Answer:** C          **GT:** A

For B, the description is self-contradictory; For C, window is not located at the center. For D, these objects are not located at the center.

Figure 14: Visualization of Object-Scene Spatial Relationship

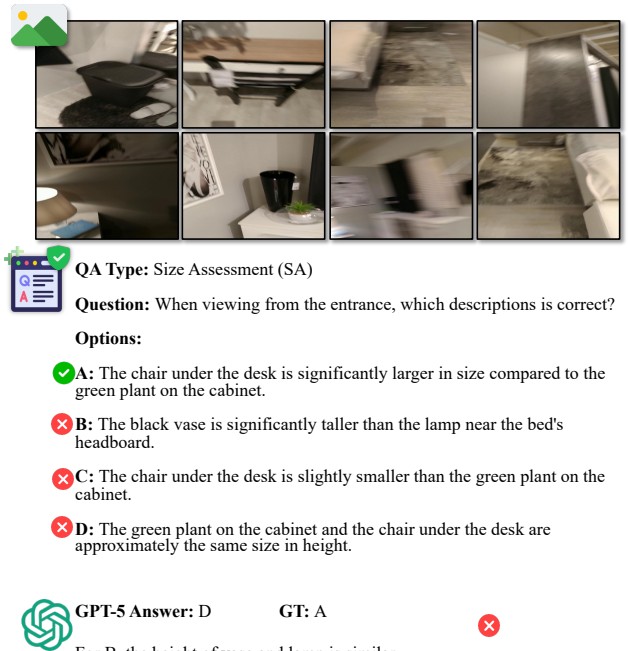

**QA Type:** Size Assessment (SA)

**Question:** When viewing from the entrance, which descriptions is correct?

**Options:**

**A:** The chair under the desk is significantly larger in size compared to the green plant on the cabinet.

**B:** The black vase is significantly taller than the lamp near the bed's headboard.

**C:** The chair under the desk is slightly smaller than the green plant on the cabinet.

**D:** The green plant on the cabinet and the chair under the desk are approximately the same size in height.

**GPT-5 Answer:** D          **GT:** A

For B, the height of vase and lamp is similar.

Figure 15: Visualization of Size Assessment

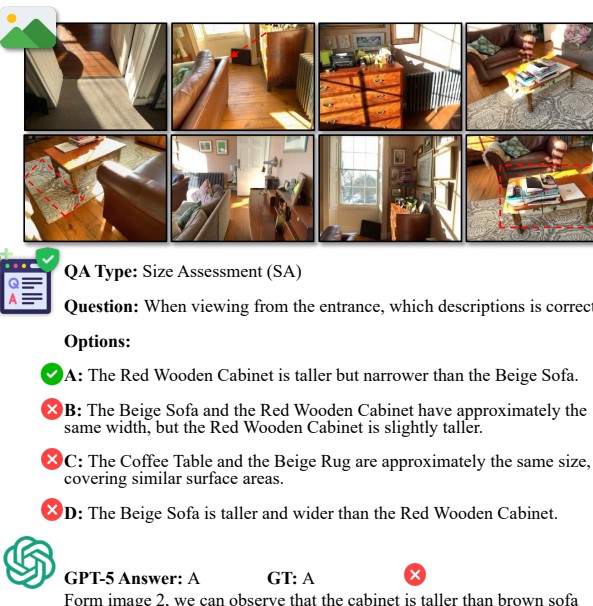

**QA Type:** Size Assessment (SA)

**Question:** When viewing from the entrance, which descriptions is correct?

**Options:**

✅**A:** The Red Wooden Cabinet is taller but narrower than the Beige Sofa.

❌**B:** The Beige Sofa and the Red Wooden Cabinet have approximately the same width, but the Red Wooden Cabinet is slightly taller.

❌**C:** The Coffee Table and the Beige Rug are approximately the same size, covering similar surface areas.

❌**D:** The Beige Sofa is taller and wider than the Red Wooden Cabinet.

**GPT-5 Answer:** A     **GT:** A     ❌

Form image 2, we can observe that the cabinet is taller than brown sofa whose height and width are similar to the beige sofa.

Figure 16: Visualization of Size Assessment

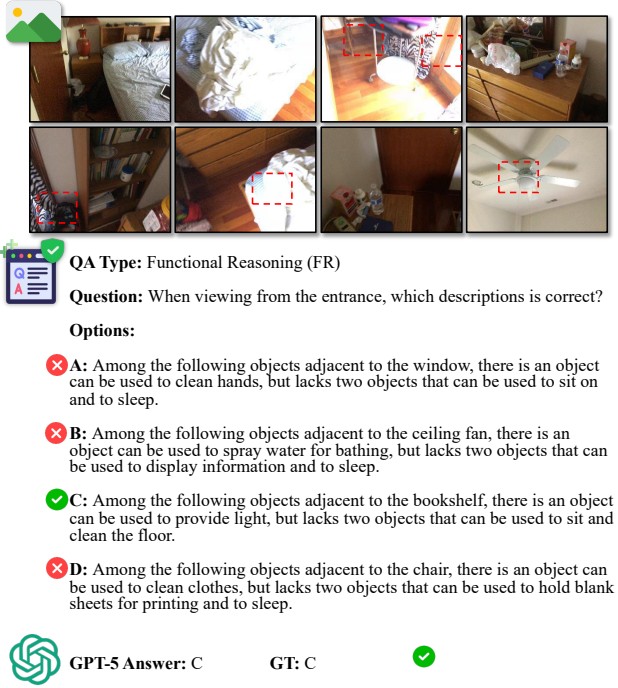

**QA Type:** Functional Reasoning (FR)

**Question:** When viewing from the entrance, which descriptions is correct?

**Options:**

❌**A:** Among the following objects adjacent to the window, there is an object can be used to clean hands, but lacks two objects that can be used to sit on and to sleep.

❌**B:** Among the following objects adjacent to the ceiling fan, there is an object can be used to spray water for bathing, but lacks two objects that can be used to display information and to sleep.

✅**C:** Among the following objects adjacent to the bookshelf, there is an object can be used to provide light, but lacks two objects that can be used to sit and clean the floor.

❌**D:** Among the following objects adjacent to the chair, there is an object can be used to clean clothes, but lacks two objects that can be used to hold blank sheets for printing and to sleep.

**GPT-5 Answer:** C     **GT:** C     ✅

For C, curtain shows that window/door is near shelf, and it can provide light. The bed is not adjacent to the shelf, thus there is no object can be used to sit.

Figure 17: Visualization of Functional Reasoning

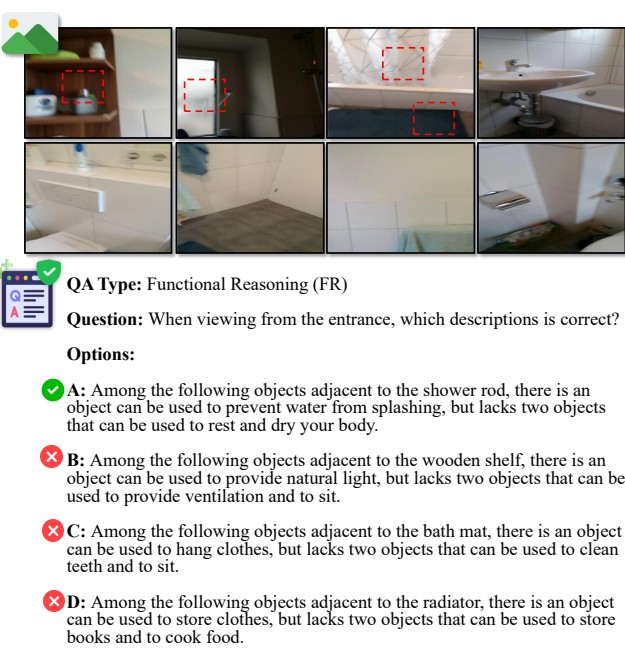

**QA Type:** Functional Reasoning (FR)

**Question:** When viewing from the entrance, which descriptions is correct?

**Options:**

**A:** Among the following objects adjacent to the shower rod, there is an object can be used to prevent water from splashing, but lacks two objects that can be used to rest and dry your body.

**B:** Among the following objects adjacent to the wooden shelf, there is an object can be used to provide natural light, but lacks two objects that can be used to provide ventilation and to sit.

**C:** Among the following objects adjacent to the bath mat, there is an object can be used to hang clothes, but lacks two objects that can be used to clean teeth and to sit.

**D:** Among the following objects adjacent to the radiator, there is an object can be used to store clothes, but lacks two objects that can be used to store books and to cook food.

**GPT-5 Answer: B**    **GT: A**

Figure 18: Visualization of Functional Reasoning

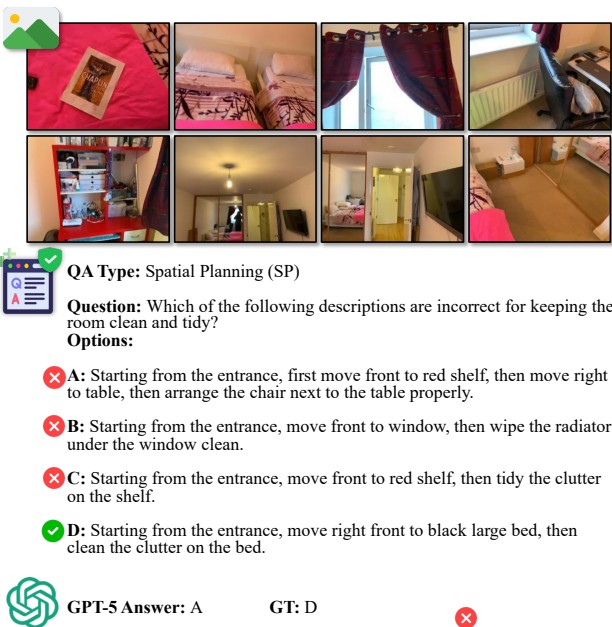

**QA Type:** Spatial Planning (SP)

**Question:** Which of the following descriptions are incorrect for keeping the room clean and tidy?

**Options:**

**A:** Starting from the entrance, first move front to red shelf, then move right to table, then arrange the chair next to the table properly.

**B:** Starting from the entrance, move front to window, then wipe the radiator under the window clean.

**C:** Starting from the entrance, move front to red shelf, then tidy the clutter on the shelf.

**D:** Starting from the entrance, move right front to black large bed, then clean the clutter on the bed.

**GPT-5 Answer:** A    **GT:** D

For D, the bed is obviously not black. Other options are correct.

Figure 19: Visualization of Spatial Planning

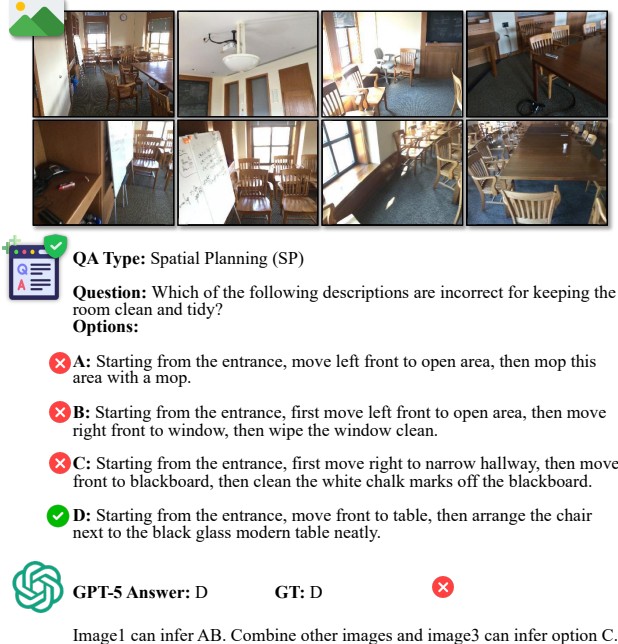

**QA Type:** Spatial Planning (SP)

**Question:** Which of the following descriptions are incorrect for keeping the room clean and tidy?
**Options:**

❌ **A:** Starting from the entrance, move left front to open area, then mop this area with a mop.

❌ **B:** Starting from the entrance, first move left front to open area, then move right front to window, then wipe the window clean.

❌ **C:** Starting from the entrance, first move right to narrow hallway, then move front to blackboard, then clean the white chalk marks off the blackboard.

✅ **D:** Starting from the entrance, move front to table, then arrange the chair next to the black glass modern table neatly.

**GPT-5 Answer:** D          **GT:** D          ❌

Image1 can infer AB. Combine other images and image3 can infer option C.

Figure 20: Visualization of Spatial Planning

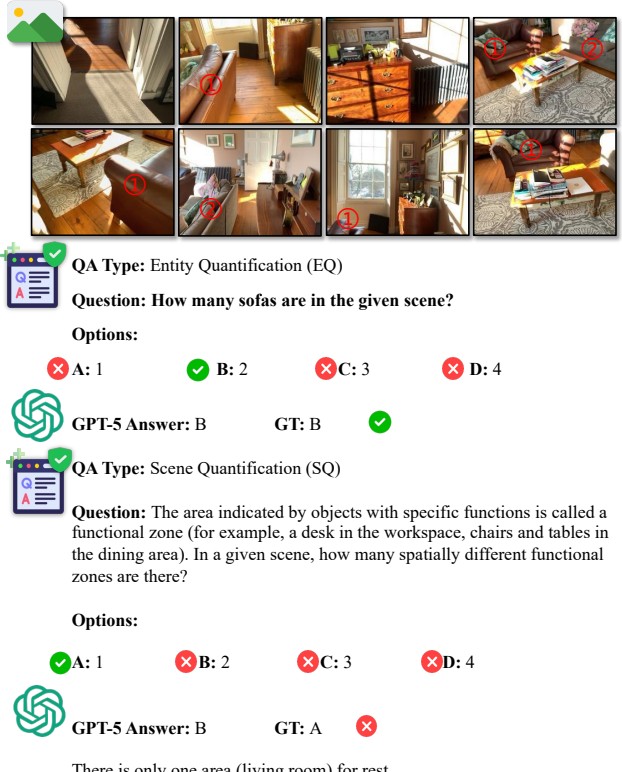

**QA Type:** Entity Quantification (EQ)

**Question: How many sofas are in the given scene?**

**Options:**

❌ **A:** 1          ✅ **B:** 2          ❌ **C:** 3          ❌ **D:** 4

**GPT-5 Answer:** B          **GT:** B          ✅

**QA Type:** Scene Quantification (SQ)

**Question:** The area indicated by objects with specific functions is called a functional zone (for example, a desk in the workspace, chairs and tables in the dining area). In a given scene, how many spatially different functional zones are there?

**Options:**

✅ **A:** 1          ❌ **B:** 2          ❌ **C:** 3          ❌ **D:** 4

**GPT-5 Answer:** B          **GT:** A          ❌

There is only one area (living room) for rest.

Figure 21: Visualization of Entity Quantification and Scene Quantification

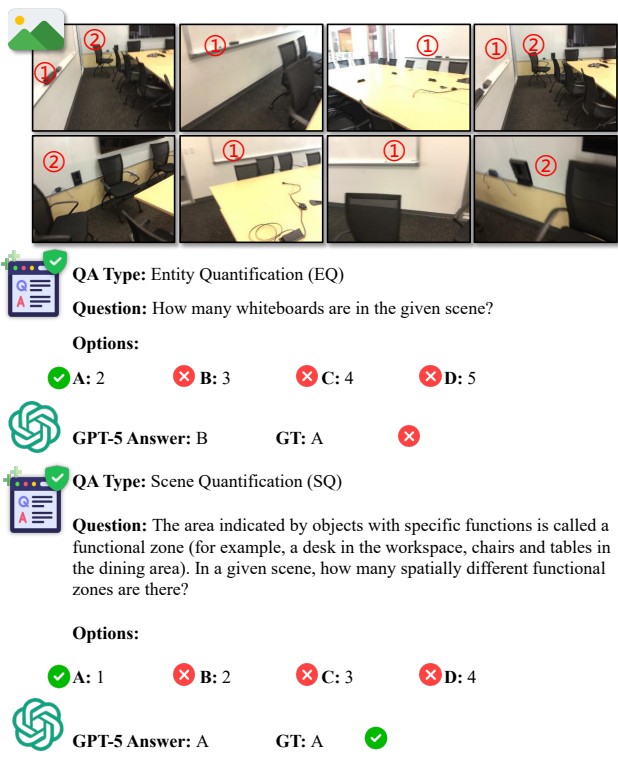

**QA Type:** Entity Quantification (EQ)

**Question:** How many whiteboards are in the given scene?

**Options:**

✅ **A:** 2 ❌ **B:** 3 ❌ **C:** 4 ❌ **D:** 5

**GPT-5 Answer:** B **GT:** A ❌

**QA Type:** Scene Quantification (SQ)

**Question:** The area indicated by objects with specific functions is called a functional zone (for example, a desk in the workspace, chairs and tables in the dining area). In a given scene, how many spatially different functional zones are there?

**Options:**

✅ **A:** 1 ❌ **B:** 2 ❌ **C:** 3 ❌ **D:** 4

**GPT-5 Answer:** A **GT:** A ✅

There is only one area for working.

Figure 22: Visualization of Entity Quantification and Scene Quantification

The critical evidence comes from View 3, where books placed on the shelves unambiguously indicate that these shelves should be recognized as bookshelves. GPT-5 most likely failed to integrate this particular viewpoint, resulting in an incorrect judgment. This illustrates that insufficient cross-view fusion, rather than object recognition, is the primary cause of error in this EP example.

(2) **Analysis in OO (Object-Object Spatial Relationship):** In the Fig. 11, the model shows that it can understand spatial relations to some extent, but the main errors come from C7 (Multi-view Fusion) and C10 (Situated Observation).

To determine that option D is incorrect, the model needs to combine information from View 3 and View 4. These two viewpoints together indicate that the toilet and the sink are positioned directly next to each other. Once we mentally imagine the scene from the "facing the front side of the objects" perspective, it becomes clear that the toilet is located on the left side of the sink rather than the right.

This error reflects a failure not in object recognition but in the integration of cross-view information and the reasoning about left–right relationships under viewpoint changes, which correspond directly to C7 and C10.

(3) **Analysis in OS (Object-Scene Spatial Relationship):** For the OS task, the first example in Fig. 13 clearly shows that the model fails on C2 (Spatial Localization) together with C7 and C10. The model selects option C instead of A, which reflects several issues. First, GPT-5 exhibits a significant global spatial mislocalization because it incorrectly places the floor-to-ceiling windows at the center of the room rather than at the far end, which directly corresponds to an error in C2. Second, to correctly choose option A, the model only needs to use the viewpoint in the last image to identify the position of the entrance. Once the entrance is recognized, the overall room layout becomes straightforward to infer, which falls under C10. These mistakes are further compounded by the model's inability to fuse complementary views across the scene, which is a typical C7 failure.

In the Fig. 14, the model again makes mistakes tied to C2, C7, and C10. Options A and C describe spatial placements that are mutually inconsistent. If the model had reasoned based on the global room layout, it would have realized that although the round dining table and chairs appear on the right in View 5, they are actually located near the center when considering the full set of views. The incorrect answer indicates that the model over-relies on isolated viewpoints, fails to integrate multi-view evidence, and does not perform viewpoint-conditioned reasoning about the true center of the room.

(4) **Analysis in SA (Size Assessment):** In Fig. 15, the error comes from a straightforward mistake in C4 (Size Comparison). The model selects option D instead of A, even though the chair under the desk is clearly much larger than the small green plant on the cabinet. This indicates a failure to perform basic relative size estimation. However, such mistakes are relatively rare. As shown in the main experimental results, C4 is one of the most stable and reliable capabilities across models, and GPT-5 generally performs well on size comparison tasks.

(5) **Analysis in FR (Functional Reasoning):** In Fig. 18, GPT-5 shows a clear deficiency in C6 (Function Knowledge). Option B is purposely designed to contain a hidden contradiction. Both "providing natural light" and "providing ventilation" are typical functions of doors or windows. If such an object can provide natural light, it almost always can provide ventilation as well, which means the two functions cannot be separated in the way option B describes. GPT-5 does not detect this implicit inconsistency and incorrectly selects B.

Option A is also intentionally challenging. The idea of "resting and drying your body" is misleading because, in View 6, although one can see a towel-like object, it is not actually adjacent to the bathtub or shower area. Since the question requires adjacency, the towel does not satisfy the functional description. Correctly identifying this requires fine-grained spatial reasoning involving the relative positions of objects, which depends on C3 (Spatial Relationship).

(6) **Analysis in SP (Spatial Planning):** In Fig. 19, we select a particularly illustrative case, since the question asks which option is incorrect. Options A and D describe almost identical navigation paths, and both include actions that are fully plausible in the given environment. However, option D contains one very obvious factual error: it refers to a "black large bed," while the bed shown in the scene is clearly not black. Despite being able to follow the spatial path correctly, GPT-5 still fails

the question because it does not detect this basic visual inconsistency. This reflects a clear failure in C1 (Object Recognition).

Such an error demonstrates that GPT-5 struggles to maintain stable perception and reasoning in cluttered or multi-step environments. Even when the spatial planning sequence is correct, a failure at the most fundamental visual capability prevents the model from arriving at the right answer. This example highlights that C1 remains essential, since higher-level planning depends directly on accurate object recognition.

**(7) Analysis in EQ (Entity Quantification) and SQ (Scene Quantification):** In Fig. 21 and 22, we observe two distinct sources of error. In the EQ questions, the model's mistakes primarily stem from the limitations of C5 (Counting). Even with multiple viewpoints available, GPT-5 fails to consistently distinguish between repeated appearances of the same object and genuinely different instances. This reflects the broader challenge we observe throughout the benchmark, where counting remains one of the weakest atomic capabilities across models.

In contrast, the errors in SQ come from an insufficient understanding of C6 (Function Knowledge). Correctly determining the number of functional zones requires understanding not only what objects are present but also how they define and structure the activities within the scene. GPT-5 struggles to identify which objects are truly function-defining and which ones serve secondary roles. As a result, it misclassifies the number of functional areas even when the visual evidence is sufficient.

Across these visualizations, we examine how all ten atomic capabilities from C1 to C10 contribute to failures across the eight QA types. Each example reveals a different weakness, ranging from basic perception issues such as object recognition and counting to more complex challenges such as multi-view fusion, functional understanding, and situated spatial reasoning. These errors collectively show that current MLLMs still lack robust and reliable foundations for compositional spatial intelligence. The variety and consistency of these failure modes strongly underscore the necessity of SpaCE-10, which provides a structured and capability-aligned benchmark to expose these limitations and guide future progress in spatial reasoning for MLLMs.

## I.2 More Detailed Statistics of SpaCE-10

SpaCE-10 contains 5,122 QA pairs in total (4,152 single-answer and 970 multiple-answer), covering 811 distinct indoor scenes with 6,488 multi-view images. The eight QA types contain the following number of samples: EQ: 388, SQ: 645, SA: 785, OO: 809, OS: 654, EP: 1,013, FR: 800, SP: 80.

Using the capability–task mapping defined in Table 5, we compute the precise distribution of samples across the ten atomic spatial capabilities: C1: 5,122, C2: 4,477, C3: 2,343, C4: 785, C5: 1,033, C6: 2,230, C7: 5,122, C8: 5,122, C9: 1,893, C10: 2,343.

## I.3 Why 3D LLMs underperform 2D MLLMs?

Although 3D representations contain richer geometry, current 3D LLMs are trained on far fewer modality-language pair data compared with 2D MLLMs. LEO (Huang et al., 2023) uses roughly 28.3M 3D–text tokens, while modern 2D MLLMs (InternVL (Zhu et al., 2025), QwenVL (Bai et al., 2025)) are trained on over 10B to 2T image–text tokens, which are about $10^3$ to $10^4$ times larger than 3D LLMs.

This large data gap means that 3D LLMs lack the world knowledge, reasoning priors, instruction-following patterns, and compositional skills that 2D models acquire from massive corpora. SpaCE-10 makes this discrepancy particularly visible.

## I.4 Why we use 1024 points for LEO?

LEO's tokenizer and transformer encoder are trained strictly on 1024-point inputs. Increasing the resolution requires re-training the entire model and is not supported by the released architecture. We therefore follow the model's official configuration to ensure fair and correct evaluation.

### I.5 FUTURE EXTENTION TO OURDOOR SCENARIOS

Outdoor spatial intelligence is an essential piece of real-world applications. We also plan to extend this capability-driven framework to outdoor settings in our following iterations. For example, we conduct a survey and initially find some potential datasets that can support the next iteration of SpaCE-10. For outdoor spatial reasoning, autonomous-driving datasets such as nuScenes (Caesar et al., 2020) and Argoverse 2.0 (Wilson et al., 2023) offer large-scale multi-sensor observations (e.g., LiDAR, multi-view RGB, HD maps) that provide rich data sources for extending capabilities such as multi-view fusion, spatial localization, and forward reasoning to both 2D and 3D evaluation. For embodied outdoor navigation, models and datasets such as ViNT (Shah et al., 2023) and CoN-VOI (Sathyamoorthy et al., 2024) demonstrate the feasibility of evaluating viewpoint-dependent reasoning and long-horizon planning beyond indoor spaces. These resources can provide feasible paths for expanding SpaCE-10 beyond indoor scenes while preserving its capability-centric design, and they will guide our future iterations toward a more comprehensive evaluation of spatial intelligence across both indoor and outdoor domains.

### I.6 THE USE OF LARGE LANGUAGE MODELS

During the writing of this manuscript, LLMs are used to generate data, judge the evaluation, and slightly polish the wording and sentences, but not to design the idea or write key sections. The authors take full responsibility for all content in the manuscript, including verifying the accuracy of LLM-assisted outputs and ensuring no plagiarism or scientific misconduct.

