# OpenReview forum: "SpaCE-10: A Comprehensive Benchmark for Multimodal Large Language Models in Compositional Spatial Intelligence"
_ICLR.cc/2026/Conference — ICLR 2026 Poster_

### Official Review · Reviewer_WP4t · 2025-10-14

**Soundness:** 2
**Presentation:** 3
**Contribution:** 2
**Rating:** 4
**Confidence:** 4

**Summary:**

The paper propose a benchmard dataset(Space-10) to evaluate MLLM's spatial understanding. It contain 5K QA for 811 real indoor scenes. The authors define 10 spatial capabilities focusing on different aspect of sptial understanding like counting, localization etc. For dataset constuction, the authors utilize multi-stage of manual and automated approaches to refine the quality. The benchmark is then used to evaluate various open and close source MLLM.

**Strengths:**

1. The author conducted extensive experiments on different MLLM on the proposed dataset.
2. The paper is well written and easy to follow. Figures look nice.
3. Evaluating on models compositional spatial capabilities is interesting and can help to understand the MLLM's capabilities.

**Weaknesses:**

1. The use of abbreviation and classes number is a bit complicated and hard to read.
2. The proposed division into ten classes is not entirely convincing, and describing them as atomic seems somewhat overstated. For instance, Classes C1, C7, and C8 all exhibit 100% coverage and are included across all eight compositional capabilities, effectively making them equivalent to the overall average. This raises the question of whether defining them as separate classes provides any meaningful distinction. Similarly, Classes C3 and C10 consistently co-occur, they are either both present or both absent within the same tasks, suggesting that none of the classes are truly independent. Given this strong interdependence, referring to them as atomic does not seem appropriate.

**Questions:**

Please see the weakness above.

---

> ### Author Response · Authors · 2025-11-23
> **Response to WP4t**
>
> We sincerely thank you for the careful reading of our paper and for raising two clear and critical questions, which are important to SpaCE-10’s presentation and the conceptual clarity of our capability formulation. Below, we briefly summarize our responses.
>
> **Q1:** The use of abbreviation and classes number is a bit complicated and hard to read.
>
> **A1:** For Q1, we understand the reviewer’s concern regarding the abbreviation and notation system. In the revision, we have added the full terms in potentially confusing sections, especially in experiments (e.g., FR for Functional Reasoning and C1 for Object Recognition), to make the descriptions easier to follow.

---

> ### Author Response · Authors · 2025-11-23
> **Response to WP4t-2**
>
> **Q2:** The proposed division into ten classes is not entirely convincing, and describing them as atomic seems somewhat overstated. For instance, Classes C1, C7, and C8 all exhibit 100% coverage and are included across all eight compositional capabilities, effectively making them equivalent to the overall average. This raises the question of whether defining them as separate classes provides any meaningful distinction. Similarly, Classes C3 and C10 consistently co-occur, they are either both present or both absent within the same tasks, suggesting that none of the classes are truly independent. Given this strong interdependence, referring to them as atomic does not seem appropriate.
>
>
> **A2:**
> **1. Why are the 10 atomic spatial capabilities meaningful and not arbitrarily designed?**
>
> Thanks for your insightful question. We’d like to first clarify that our capability taxonomy is not arbitrarily designed. Instead, it stems from well-established research lines in spatial intelligence, as shown in our appendix:
>
> “Initially, a beginning reference for our design is Sparkle[1], which frames 2D spatial intelligence around three abilities—Direction, Distance, and Location. In SpaCE-10, the direction and distance are split into C3 (Spatial Relationship). Since we believe that the concept of ‘what and where’ is one of the most fundamental abilities for real spatial intelligence, location is separated and mapped to the C2 (Spatial Location). Next, early studies [2,3] mainly emphasize C4 (Size Comparison) and simple positional cues in clean images. Then, as evaluation expands from single-scene 3D point-cloud QA [4-11] to multi-view 2D imagery [12-16], the focus shifts from single-view C3 (Spatial Relationship) and C5 (Counting) within one scene to cross-view consistency captured by C7 (Multi-view Fusion).
> In parallel, SQA [6] introduces viewpoint conditioning, aligning with C10 (Situated Observation). Nowadays, with the rise of Embodied Intelligence and Reasoning MLLMs, based on these previous excellent works, we further propose to incorporate C6 (Function Knowledge), C8 (Forward Thinking), and C9 (Reverse Reasoning) into this atomic capability pool for manipulating
> basic knowledge and reasoning ability examination.”
>
> **2. Why ‘atomic’ does not mean independent?**
>
> Second, when we refer to these capabilities as 'atomic', we do not mean that they must be mutually independent. Instead, it means that each capability cannot be further decomposed. Also, in compositional spatial reasoning, multiple atomic abilities naturally co-occur because complex spatial tasks inherently require diverse capabilities to complete.
>
> **3. Interpreting the full-coverage abilities (C1, C7, C8)**
>
> Regarding that C1, C7, and C8 have 100% coverage, this does not indicate a flaw in our taxonomy. Instead, it reflects the fact that these three abilities are essential prerequisites for almost all spatial tasks. Without C1 (object recognition), a model cannot understand any object in a scene; without C7 (multi-view fusion), it cannot integrate complementary visual cues; and without basic C8 (forward thinking), it cannot complete any thinking process. Their high coverage reflects the structure of spatial tasks themselves rather than a limitation in our categorization. Separating them may make the questions overly simple and reduce evaluation's meaning.
>
> **4. Why C3 (Spatial Relationship) and C10 (Situated Observation) co-occur and how they differ？**
>
> We would like to emphasize that **this co-occurrence is a natural property of spatial reasoning rather than that they are indistinguishable**. In multiple-view observation, spatial relations inherently depend on specific viewpoints. For example, the statement “A is to the left of B” is only meaningful under a particular view; without this view, the relation will be unclear and undefined. Therefore, C3 and C10 frequently appear together due to a viewpoint is required to interpret spatial relations.
>
> However, C3 can appear alone in single-view QAs because the model only needs to observe within a fixed viewpoint. There is no need for C10 when the view is already fixed. But such cases are usually simple and probably can not provide a sufficient test for current powerful MLLMs.
>
> **5. Our overall design philosophy**
>
> Finally, as the first benchmark for compositional spatial intelligence, our goal is not to force complete independence among atomic capabilities, but to attempt to structure them in a way that reveals MLLMs' spatial intelligence. Our experiments in the paper also show clear differences across these capabilities, demonstrating the initial effect of this disentanglement.
>
> We also thank the reviewer for this insightful question. We fully agree that exploring task formulations where atomic capabilities can be split more explicitly is an interesting direction. In future work, we will further make more investigate how to further split these atomic spatial capabilities while maintaining question quality.

---

> > ### Author Response · Authors · 2025-11-23
> > **Response to WP4t-3**
> >
> > ### **Reference:**
> >
> > [1] Sparkle: Mastering basic spatial capabilities in vision language models elicits generalization to composite spatial reasoning.
> >
> > [2] What’s” up” with vision-language models? investigating their struggle with spatial reasoning.
> >
> > [3] Are elephants bigger than butterflies? reasoning about sizes of objects.
> >
> > [4] Scanqa: 3d question answering for spatial scene understanding.
> >
> > [5] Sqa3d: Situated question answering in 3d scenes.
> >
> > [6] 3d question answering.
> >
> > [7] Toward explainable 3d grounded visual question answering: A new benchmark and strong baseline.
> >
> > [8] 3d-llm: Injecting the 3d world into large language models.
> >
> > [9] 3d-grand: A million-scale dataset for 3d-llms with better grounding and less hallucination.
> >
> > [10] Mmscan: A multi-modal 3d scene dataset with hierarchical grounded language annotations.
> >
> > [11] Unveiling the mist over 3d vision-language understanding: Objectcentric evaluation with chain-of-analysis.
> >
> > [12] 3dsrbench: A comprehensive 3d spatial reasoning benchmark.
> >
> > [13] Multi-modal situated reasoning in 3d scenes.
> >
> > [14] Thinking in space: How multimodal large language models see, remember, and recall spaces.
> >
> > [15] Omnispatial: Towards comprehensive spatial reasoning benchmark for vision language models.
> >
> > [16] Spatialrgpt: Grounded spatial reasoning in vision-language models.

---

> > ### Comment · Reviewer_WP4t · 2025-11-23
> > **Thanks for the rebuttal**
> >
> > Thanks again for the rebuttal. The response addresses part of my concern, but I still have one remaining question that I hope the authors can elaborate on.
> >
> > While I understand that the term atomic does not necessarily imply independence, most of the proposed high-level understanding also do not seem “non-decomposable.” To clarify, my concern is not about the linguistic definition of atomic. Rather, my initial understanding was that one of the main strengths of the paper, and what differentiates it from prior work, is the ability to perform fine-grained, isolated evaluation of specific spatial capabilities.
> > However, if certain capabilities are inherently inseparable or appear in all questions, then they seem somewhat redundant from an evaluation perspective, because they do not provide additional discriminative power when comparing models (though I am not suggesting these abilities are unimportant).
> >
> > Therefore, I would appreciate clarification on the contribution of the proposed benchmark, especially regarding:
> > Which capabilities in SpaCE-10 can be evaluated only using your benchmark and are not reliably tested by previous benchmarks? I assume that C1, C7, and C8 may already be implicitly evaluated in existing work, since these abilities seem to be essential prerequisites for almost all spatial tasks.
> >
> > Additionally, although the benchmark contains 5k QA pairs in total, the number of samples per capability appears relatively small, which may limit the reliability of capability-specific evaluation. Would be nice if the authors can clarify these, thanks a lot.

---

> > > ### Author Response · Authors · 2025-11-25
> > > **Thanks for your review！**
> > >
> > > **Q1:** Which capabilities in SpaCE-10 can be evaluated only using your benchmark and are not reliably tested by previous benchmarks?
> > >
> > > **A1:**
> > > We thank the reviewer for this quite thoughtful question and the kind attitude in engaging with our work. This question is actually considered during curating SpaCE-10. Before clarifying which capabilities are uniquely enabled by our benchmark, we would like to briefly restate our motivations and contributions.
> > >
> > > 1. **We are the first to explicitly formalize a capability pool for spatial intelligence** and to use it as the principle of the whole work.
> > >
> > > 2. **All QA types are designed from the capability perspective**, so each question intentionally combines multiple capabilities for compositional spatial intelligence evaluation.
> > >
> > > 3. **We map task-level accuracy into capability-level accuracy**, providing the community with the first diagnosis of spatial capabilities of existing MLLMs.
> > >
> > > As our previous response, the 10 capabilities are inspired by prior works, which makes it difficult to completely avoid overlap with others. Our goal, however, is to encourage the community to take *spatial capabilities* into consideration to enable more reasonable and specific dataset construction and model training, rather than relying on simply data mixing.
> > >
> > >
> > > **Existing benchmark coverage and missing capabilities.**
> > >
> > > The early forms of spatial QA benchmarks originate from the 3D area, with representative works including ScanQA [1] and SQA3D [2].
> > >
> > >
> > > **ScanQA** primarily evaluates C2 (Spatial Localization) and C3 (Spatial Relationship) using global point clouds. It does not reliably test:
> > >
> > > 1.**C7 (Multi-view Fusion):** inputs lack viewpoint variation.
> > >
> > > 2.**C10 (Situated Observation):** questions do not specify viewpoints, making left/right relations ambiguous.
> > >
> > > 3.**C4 (Size Comparison), C5 (Counting), C6 (Function Knowledge), C9 (Reverse Reasoning):** these capabilities are rarely covered.
> > >
> > >
> > > **SQA3D** improves C10 through explicit viewpoint descriptions (e.g., ‘sitting at the edge of the bed and facing the couch’ ) but still lacks C4, C5, C6, C7, C9, and most questions remain simple combinations of C1–C3.
> > >
> > > As the paradigm evolved, **MSQA [3]** began to combine 3D point clouds with interleaved text and image. Based on the format of SQA3D, MSQA replaces the **word** of objects in the questions with the **image**. Despite this broadening of modalities, its capability coverage is similar to SQA3D.
> > >
> > > Beyond 3D research, earlier 2D attempts explore only limited capability types such as size comparison [4]. VSI [5] significantly expands coverage in 2D, covering C1, C2, C3, C4, C5, C7, C8, and C10, but still does not reliably test:
> > >
> > > 1. **C6 (Function Knowledge)**, which is crucial for affordance understanding and tool usage of manipulation tasks;
> > > 2. **C9 (Reverse Reasoning)**, which is essential for causal and planning-based spatial inference.
> > >
> > > Most importantly, none of these benchmarks investigates spatial intelligence from the perspective of spatial capability, and this is precisely the motivation behind SpaCE-10.
> > >
> > >
> > >
> > > **Q2:** Additionally, although the benchmark contains 5k QA pairs in total, the number of samples per capability appears relatively small, which may limit the reliability of capability-specific evaluation.
> > >
> > > **A2:** We appreciate the reviewer’s observation and fully agree that increasing the per-capability sample size will further strengthen statistical reliability. We therefore plan to expand the dataset in the next version to provide denser coverage across all capabilities.

---

> > > > ### Comment · Reviewer_WP4t · 2025-11-25
> > > > **Thanks for the explanation !**
> > > >
> > > > Thanks for the explanation. I will adjust the score to 6.

---

> > > > > ### Author Response · Authors · 2025-11-25
> > > > >
> > > > > We sincerely appreciate your effort, patience, and professional attitude throughout the entire rebuttal process. Thank you very much for your supportive feedback.

---

### Official Review · Reviewer_D8Pf · 2025-10-27

**Soundness:** 2
**Presentation:** 3
**Contribution:** 2
**Rating:** 4
**Confidence:** 2

**Summary:**

The paper presents SpaCE-10, a comprehensive benchmark for evaluating compositional spatial intelligence in multimodal large language models. It defines 10 atomic spatial capabilities and systematically combines them into 8 compositional QA types, built from 811 real indoor scenes with 5k+ question–answer pairs across both 2D image and 3D point-cloud settings.

**Strengths:**

1. The paper clearly explains ten basic spatial skills and combines them into eight question types. This setup makes it easier to see which abilities the models are good or bad at, instead of only looking at overall accuracy.
2. The data collection process is well organized, mixing automated generation with human checking to keep questions accurate and varied. The experiments on about 50 models give useful insights.

**Weaknesses:**

1. Overall, the benchmark is limited to indoor scenes, which narrows its scope. Real-world spatial intelligence also involves outdoor and embodied settings, for example, navigation and perception in autonomous driving or robotics.
2. Fixing inputs to 8 images may limit multi-view reasoning. What is the performance when the number of views grows?
3. MCQ-only setup: This misses tasks that need precise outputs (e.g., 3D grounding with (x,y,z) coordinates, path planning). What is the current status of MLLM performance on these tasks?
4. Intuitively, reasoning directly in 3D should work better, but in this benchmark the 3D LLMs perform worse than the 2D LLMs. Would denser point clouds help? 1,024 points seem too sparse and may not capture the whole scene. Also, would it be possible to use the same 8 images with a feed-forward reconstructor (e.g., VGGT) to build a point cloud, then feed that into the 3D MLLM?

**Questions:**

See weakness

---

> ### Author Response · Authors · 2025-11-23
> **Response to D8Pf**
>
> We sincerely thank the reviewer for the time and effort devoted to reading our submission and for raising these insightful questions. Below, we provide our detailed responses.
>
> **Q1:** Overall, the benchmark is limited to indoor scenes, which narrows its scope. Real-world spatial intelligence also involves outdoor and embodied settings, for example, navigation and perception in autonomous driving or robotics.
>
> **A1:** We thank the reviewer for the insightful comment. We fully agree with the importance of the outdoor scenes, and we are glad to discuss how to further expand SpaCE-10 to diverse open-world settings. But before this, we hope to briefly explain our motivation of why we focus on indoor scenes.
>
> **Meaning and challenge of indoor scenes**
>
> Our concentration of indoor scenes stems from considerations of how spatial intelligence can be applied in real-world settings. Compared with fully open-world outdoor environments, indoor spaces are enclosed but also **meaningful**, and they still contain many spatial challenges that remain **unsolved**.
>
> From early 3D spatial works, ScanQA [1], SQA3D [2], 3D LLMs [3,4,5], and the recently released VSI [6], BEHAVIOR-1K [7] benchmarks by Feifei Li all focus on indoor tasks and share a common motivation: indoor environments are directly tied to humans’ daily life applications, especially in our living space. Indoor scenes also provide a more feasible starting point for reliable spatial intelligence due to more controllable and scalable data collection. Despite these works and advantages, indoor scenes still face severe spatial problems: dense multi-object layouts, severe occlusion, complex geometry, and diverse viewpoint changes. Recent works such as VSI [6] further validate that current MLLMs still struggle with indoor scenes. Yet, there is little discussion on what spatial capabilities are required to overcome these challenges, and how these capabilities should be defined, measured, or combined. Thus, we make the first attempt to introduce SpaCE-10, a capability-driven benchmark that evaluates compositional spatial intelligence. Given this context, our benchmark focuses on indoor environments.
>
> **Toward outdoor extensions**
>
> Nevertheless, outdoor spatial intelligence remains an essential piece of real-world applications. We also plan to extend this capability-driven framework to outdoor settings in our following iterations. For example, we conduct a survey and initially find some potential datasets that can support the next iteration of SpaCE-10. For outdoor spatial reasoning, autonomous-driving datasets such as nuScenes [8] and Argoverse 2.0 [9] offer large-scale multi-sensor observations (e.g., LiDAR, multi-view RGB, HD maps) that provide rich data sources for extending capabilities such as multi-view fusion, spatial localization, and forward reasoning to both 2D and 3D evaluation. For embodied outdoor navigation, models and datasets such as ViNT [10] and CoNVOI [11] demonstrate the feasibility of evaluating viewpoint-dependent reasoning and long-horizon planning beyond indoor spaces. These resources can provide feasible paths for expanding SpaCE-10 beyond indoor scenes while preserving its capability-centric design, and they will guide our future iterations toward a more comprehensive evaluation of spatial intelligence across both indoor and outdoor domains.

---

> ### Author Response · Authors · 2025-11-23
> **Response to D8Pf-2**
>
> **Q2:** Fixing inputs to 8 images may limit multi-view reasoning. What is the performance when the number of views grows?
>
>
> | Num |   EQ  |  SQ  |   SA  |   OO  |   OS  |   EP  |   FR  |   SP  | Avg. |
> |-----|-------|------|-------|-------|-------|-------|-------|-------|---------|
> | 1   | 23.3  | 25.1 | 39.6  | 37.0  | 33.2  | 16.1  | 37.2  | 16.3  | 28.5  |
> | 2   | 25.3  | 25.6 | 42.5  | 42.7  | 34.3  | 20.3  | 47.3  | 22.7  | 32.6 |
> | 4   | 33.6  | 30.4 | 41.4  | 48.8  | 37.6  | 21.1  | 55.3  | 30.9  | 37.4 |
> | 8   | 36.6  | 29.5 | 42.9  | 51.7  | 34.5  | 26.6  | 60.6  | 37.5  | 40.0 |
> | 16  | 30.0  | 30.6 | 44.3  | 55.4  | 38.3  | 30.7  | 64.4  | 36.2  | 41.2 |
>
>
>
> **A2:** We thank the reviewer for this important question. We analyze the effect of the number of views by evaluating InternVL3-8B [23] with 1, 2, 4, 8, and 16 input frames.
>
> We observe substantial gains when increasing the number of views from 1 to 8 (+11.5%), confirming the importance of multi-view information for compositional spatial intelligence. From 8 to 16 views, accuracy improves only slightly (+1.2%), showing lower marginal benefit between performance and input images.  Below, we provide a more detailed analysis of some QAs.
>
> For EQ (Entity Quantification), performance rises from 8 to 16 views but drops clearly by 6.6% (36.6% to 30.0%). This reflects a core limitation of current MLLMs: with dense viewpoints, they struggle to maintain object identity consistency and tend to count the same object multiple times. EQ exposes this instability because it depends on reliable multi-view fusion (C7) rather than simply accumulating more frames.
>
> By contrast, **OO (Object-Object Spatial Relationship)**, **OS (Object-Scene Spatial Relationship)**, **EP (Entity Presence)**, and **FR (Functional Reasoning)** all improve over 3% from 8 to 16 views.
>
> (1) **OO** and **OS** gain from additional viewpoints because both tasks depend on clear spatial relationships and object geometry. More views can reduce occlusion and reveal spatial information that is missing under sparse observations.
>
> (2) **EP** also improves, as wider viewpoint coverage exposes objects that may be partially occluded or located near scene boundaries, strengthening object-level appearance without involving counting.
>
> (3) **FR** benefits as well. More views reveal a larger portion of each central object’s surrounding area, making the local layout clearer and helping the model judge nearby objects’ functions.
>
> For **SP (Spatial Planning)**, performance slightly decreases when increasing from 8 to 16 views. SP relies heavily on cross-view consistency, and additional viewpoints may introduce redundant or inconsistent layout information. This highlights the instability of current multi-view reasoning mechanisms in today's MLLMs.
>
>
> Overall, increasing the number of views improves performance, but it is not the main problem, especially when the numbers reach 16. This reveals that the core challenge lies not in the number of available views but in how well MLLMs interpret and integrate the spatial information already presented in the given observations. Simply adding more views cannot fully compensate for inconsistent multi-view reasoning.
>
>
> These findings also potentially validate the quality and the intention of SpaCE-10: the benchmark evaluates whether models can truly understand spatial information and combine diverse spatial capabilities, rather than rely on simple viewpoint accumulation.

---

> ### Author Response · Authors · 2025-11-23
> **Response to D8Pf-3**
>
> **Q3:** MCQ-only setup: This misses tasks that need precise outputs (e.g., 3D grounding with (x,y,z) coordinates, path planning). What is the current status of MLLM performance on these tasks?
>
> **A3:** Thank you for this valuable question. We fully agree that tasks requiring precise outputs, such as (x, y, z) grounding or path planning, represent some of the most advanced forms of spatial intelligence. These tasks demand extremely fine-grained geometric understanding, and they will remain essential for the long-term progress of the field. Below, we discuss the advantages and limitations of both formats and explain why SpaCE-10 adopts an MCQ formulation at this stage.
>
> **1. MCQ provides a unified and efficient evaluation metric.**
>
> Since SpaCE-10 is the first benchmark for atomic spatial capabilities, we need a task format that stays consistent across tasks and atomic spatial capabilities. MCQ provides a stable text-based interface that all 2D and 3D MLLMs can handle and allows efficient, accuracy-based evaluation, making large-scale comparison across nearly fifty models feasible. By contrast, continuous numeric outputs require task-specific metrics, for example, Acc@k for coordinate grounding [12], ADE/FDE for trajectory prediction [13], or Chamfer Distance for 3D reconstruction [14], which makes it difficult to maintain a unified and fair comparison across models.
>
> **2. Better interpretability with MCQ**
>
> Second, the MCQ adopted by lots of works [6,15] offers a controlled answer space that makes model behavior more interpretable. Because each distractor can be designed to represent a plausible but incorrect spatial outcome, MCQ enables clearer identification of error patterns, an approach that prior visual reasoning and multimodal QA benchmarks adopt frequently [16]. In contrast, continuous numeric outputs provide only a raw value, making it much harder to trace or categorize the underlying spatial reasoning errors.
>
> **3. Challenges in using precise numeric outputs**
>
> Third, for current general-purpose MLLMs that are not trained on specialized data for precise spatial outputs, numeric questions introduce additional noise and instability that interfere with measuring spatial capabilities. VSI [6] validates that humans who already understand the spatial information of scenes still struggle to pass strict numeric tests. This suggests that a precise output question may blur real spatial understanding evaluation due to the lack of numeric output formats. Moreover, coordinate-style questions raise several fundamental design challenges: determining which camera defines the coordinate origin, choosing between absolute and relative scales, and ensuring that all viewpoints share a consistent coordinate system. Therefore, we view precise numeric prediction as more appropriate for later stages of evaluation.
>
> **4. Status and Future Improvement**
>
> For the status of MLLM development on these tasks, recent 3D works such as BEACON3D [17] start to explore 3D grounding for dedicated 3D models under carefully specified coordinate conventions, but analogous 3D grounding from 2D images for general MLLMs is still largely underexplored. Nevertheless, we fully agree that continuous prediction tasks are a valuable direction. As models develop more reliable numeric output behavior, we will also plan to explore how precise spatial predictions can be combined with capability-driven analysis to achieve a more comprehensive assessment.

---

> ### Author Response · Authors · 2025-11-23
> **Response to D8Pf-4**
>
> **Q4:** Intuitively, reasoning directly in 3D should work better, but in this benchmark the 3D LLMs perform worse than the 2D LLMs. Would denser point clouds help? 1,024 points seem too sparse and may not capture the whole scene. Also, would it be possible to use the same 8 images with a feed-forward reconstructor (e.g., VGGT) to build a point cloud, then feed that into the 3D MLLM?
>
>
> **A4:** We thank the reviewer for the insightful questions. We agree that denser point clouds could help in principle. However, the performance gap observed in SpaCE-10 mainly reflects current limitations of 3D LLMs, not the sparsity of our inputs.
>
> **1. Why 3D LLMs underperform 2D MLLMs?**
>
> Although 3D representations contain richer geometry, current 3D LLMs are trained on far fewer modality-language pair data compared with 2D MLLMs. LEO[4] uses roughly **28.3M** 3D–text tokens, while modern 2D MLLMs (InternVL [23], QwenVL [24]) are trained on **over 10 B to 2 T** image–text tokens, which are about **10³ to 10⁴** times larger than 3D LLMs.
>
> This large data gap means that 3D LLMs lack the world knowledge, reasoning priors, instruction-following patterns, and compositional skills that 2D models acquire from massive corpora. SpaCE-10 makes this discrepancy particularly visible.
>
>
> **2. Why 1024 points?**
>
> LEO’s tokenizer and transformer encoder are trained strictly on **1024-point inputs**. Increasing the resolution requires re-training the entire model and is not supported by the released architecture. We therefore follow the model’s official configuration to ensure fair and correct evaluation.
>
> Thus, the performance gap is not caused by insufficient point cloud, but by the structural constraints of current 3D LLMs.
>
>
> **3. Would VGGT [18]-like reconstruction help?**
>
> VGGT can produce denser, smoother 3D reconstructions, but this does not benefit our evaluation:
>
> (1) **Model Capacity**: the 3D LLM cannot ingest >1024 points. Additional detail is simply discarded.
> (2) **Data Source Quality**: SpaCE-10 already uses high-fidelity scenes (ScanNet [19], ScanNet++ [20], ARKitScene [21], 3RScan[22]) and they contain official dense cloud point. The 1024-point clouds are sampled only to match the model’s architecture, not because the raw scenes are sparse.
>
> Therefore, denser reconstructions do not address the real bottleneck.
>
> **Conclusion**
>
> Through this fair comparison and in-depth discussion between 2D and 3D models, SpaCE-10 also reveals two concrete improvement directions for future 3D LLMs:
> (1) Scaling up 3D-language alignment toward the breadth and diversity currently seen in large 2D MLLMs, which is essential for building stronger reasoning priors.
> (2) Developing more efficient and flexible 3D tokenizers that can support higher-density point cloud inputs is very urgent.
>
> We appreciate the reviewer’s insightful findings and have clarified these discussions in the appendix of the revision.
>
> ### **Reference:**
>
> [1]ScanQA: 3D Question Answering for Spatial Scene Understanding.
>
> [2]SQA3D: Situated Question Answering in 3D Scenes.
>
> [3]3D-LLM: Injecting the 3D World into Large Language Models.
>
> [4]An Embodied Generalist Agent in 3D World.
>
> [5]LLaVA-3D: A Simple yet Effective Pathway to Empowering LMMs with 3D-awareness.
>
> [6]Thinking in Space: How Multimodal Large Language Models See, Remember, and Recall Spaces.
>
> [7]BEHAVIOR-1K: A Human-Centered, Embodied AI Benchmark with 1,000 Everyday Activities and Realistic Simulation.
>
> [8]nuScenes: A multimodal dataset for autonomous driving.
>
> [9]Argoverse 2: Next Generation Datasets for Self-Driving Perception and Forecasting.
>
> [10]ViNT: A Foundation Model for Visual Navigation.
>
> [11]CoNVOI: Context-aware Navigation using Vision Language Models in Outdoor and Indoor Environments.
>
> [12]ScanRefer: 3D Object Localization in RGB-D Scans using Natural Language.
>
> [13]From Goals, Waypoints & Paths to Long-term Human Trajectory Forecasting.
>
> [14]Density-aware Chamfer Distance as a Comprehensive Metric for Point Cloud Completion.
>
> [15]OST-Bench: Evaluating the Capabilities of MLLMs in Online Spatio-temporal Scene Understanding.
>
> [16]Automated generation of challenging multiple-choice questions for vision language model evaluation.
>
> [17]Unveiling the Mist over 3D Vision-Language Understanding: Object-centric Evaluation with Chain-of-Analysis.
>
> [18]Vggt: Visual geometry grounded transformer.
>
> [19]Scannet: Richly-annotated 3d reconstructions of indoor scenes.
>
> [20]Scannet++: A high-fidelity dataset of 3d indoor scenes.
>
> [21]Arkitscenes: A diverse real-world dataset for 3d indoor scene understanding using mobile rgb-d data.
>
> [22]Rio: 3d object instance re-localization in changing indoor environments.
>
> [23]InternVL3: Exploring Advanced Training and Test-Time Recipes for Open-Source Multimodal Models.
>
> [24]Qwen2.5-VL Technical Report.

---

> ### Comment · Reviewer_D8Pf · 2025-11-24
>
> Thank you for the clarification. It’s great to hear that extending the benchmark to outdoor scenes is in your future plans. Can you include this into the future work section?
>
> Regarding the VGGT feed-forward reconstruction comment, my intention was to clarify the note on lines 308–309 (“To ensure a fair comparison…”). I initially interpreted “fair comparison” as balancing the number of input images and points which using feed-forward point clouds reconstructed from the same 8 images to ensure that both modalities provide equivalent scene information in different formats. I now understand that the 1,024 points sampling is due to the model’s fixed training setup. It would be helpful to mention this downsampling is to match the model’s training set up.
>
> Overall, I appreciate the author’s detailed explanations and I’m willing to increase my rating to 6.

---

> > ### Author Response · Authors · 2025-11-24
> > **Response to D8Pf**
> >
> > Thanks for your thoughtful comments and willingness to increase the score. Following your suggestion, we have updated the discussion of future extension to outdoor scenarios in Section I.5 of the Appendix. We truly appreciate your professional and kind feedback.

---

### Official Review · Reviewer_sqZm · 2025-10-31

**Soundness:** 3
**Presentation:** 4
**Contribution:** 4
**Rating:** 8
**Confidence:** 5

**Summary:**

This paper propose SpaCE-10 to address the limitation of existing benchmark in comprehensively assessing the ability of MLLMs to integrate multiple basic (the atomic level) spatial reasoning skills into complex, compositional tasks (the compositional level). The proposed benchmark defines a structured hierarchy, including 10 atomic spatial capabilities which are combined into 8 compositional capabilities. The authors design a hierarchical annotation pipeline to generate QA pairs, together with assistance from human experts.
Experiments across models from 1B to 200+B provide in-depth analysis, with special focus on single abilities compared with their combination. These results reveal significant shortage of MLLMs in integrated spatial intelligence.

**Strengths:**

- Significance:
  1. The addressed compositional spatial reasoning ability is indeed important, challenging and useful in real world tasks.
  2. The identified performance gap between MLLMs and humans on SpaCE-10 highlights the immediate and practical value of this benchmark in directing future model development.

- Originality: The paper offers an original and highly structured framework that 1. clearly defines 10 atomic level and 8 compositional level spatial capabilities, and 2. provides novel insights for researchers to precisely diagnose model deficiencies in 2D and 3D spatial reasoning.

- Quality:
  1. The conducted extensive evaluation of more than 50 MLLMs with diverse scales and structures validates the benchmark's utility. These results reveal common trends other than determining shortages for specific models.
  2. This work includes both data generation, quality verification, in-depth analysis and case studies, almost covering aspect required by a benchmark.

- Clarity:
  1. This paper is well motivated and demonstrated.
  2. The clear organization of spatial reasoning into atomic and compositional levels provides an intuitive and logical structure for evaluating intelligence in space.

**Weaknesses:**

1. This paper only covers indoor scenes, especially the housing scenes. While there are lots of different scenes worth investigating, such as indoor inductrial scene (in a factory), and also outdoor scenes. These could further expand the coverage and generality of the scope of this benchmark.
2. Could also provide detialed insights for future research (further discuss the cause and potential improvements for your findings), or indicate example practical tasks for potential applications.

**Questions:**

1. Please provide a detailed breakdown of the SpaCE-10 benchmark. Specifically, what is the total number of QA pairs, the number of distinct images used, and the precise distribution of samples across the 10 atomic and 8 compositional capabilities?
2. Please to the points in Weaknesses section.
3. This is not a question, just to clarify that I want to assign a rate of 9 (but only allowed to choose between 8 and 10). The reason for not assigning the highest score is stated in Weaknesses#1. I suppose this work could be further strengthened if covering more practical scenes. Overall this is a good work.

---

> ### Author Response · Authors · 2025-11-23
> **Response to sqZm**
>
> We sincerely thank you for your exceptionally positive comment. We truly appreciate the high score, the encouraging assessment of our contribution, and the thoughtful suggestions for further strengthening the SpaCE-10. We are very grateful for the time you dedicated to reviewing our work.
>
>
> **Q1:** This paper only covers indoor scenes, especially the housing scenes. While there are lots of different scenes worth investigating, such as indoor industrial scene (in a factory), and also outdoor scenes. These could further expand the coverage and generality of the scope of this benchmark.
>
>
> **A1:** We sincerely thank the reviewer for this thoughtful and constructive suggestion. We fully agree that industrial indoor environments (such as factories and warehouses) and outdoor scenes are highly meaningful for advancing spatial intelligence. Industrial settings are central to real-world deployment in logistics and manufacturing, while outdoor environments represent one of the most important and challenging directions for open-world spatial intelligence, including navigation, locomotion, and large-scale spatial planning.
>
> Based on this suggestion, we also conduct an initial survey for our further extension. For example, we initially find some potential datasets that can support the next iteration of SpaCE-10. For outdoor spatial reasoning, autonomous-driving datasets such as nuScenes [1] and Argoverse 2.0 [2] offer large-scale multi-sensor observations (e.g., LiDAR, multi-view RGB, HD maps) that provide rich data sources for extending capabilities such as multi-view fusion, spatial localization, and forward reasoning to both 2D and 3D evaluation. For embodied outdoor navigation, models and datasets such as ViNT [3] and CoNVOI [4] demonstrate the feasibility of evaluating viewpoint-dependent reasoning and long-horizon planning beyond indoor spaces.
>
> Additionally, although we have not yet identified benchmark or dataset papers dedicated specifically to industrial or factory environments, we can still leverage image resources from related areas, such as industrial anomaly detection and object detection, to extend SpaCE-10 into industry-oriented scenarios. These resources can provide feasible paths for expanding SpaCE-10 beyond indoor scenes while preserving its capability-centric design, and they will guide our future iterations toward a more comprehensive evaluation of compositional spatial intelligence across both indoor and outdoor domains.

---

> > ### Author Response · Authors · 2025-11-23
> > **Response to sqZm-2**
> >
> > **Q2:** Could also provide detailed insights for future research (further discuss the cause and potential improvements for your findings), or indicate example practical tasks for potential applications.
> >
> >
> >
> >
> > **A2:** Thank you for this excellent suggestion. We are glad to discuss more concrete insights about how the atomic spatial capability-oriented paradigm of SpaCE-10 can guide future research and connect to real-world applications.
> >
> > **1. Capability-Aware Training Paradigm:**
> > Our analysis in Fig. 4 reveals that current MLLMs show limitations not randomly, but systematically, along specific atomic capabilities. For example, MLLMs often miscount because their probability-based next-token prediction and training data are not designed for precise, object-level counting. This structure naturally suggests a capability-aware data construction and training paradigm.
> >
> > We think that future datasets can be organized or generated according to these abilities, instead of simply mixing large-scale spatial corpora to validate the scaling law. For example:
> >
> > **(1) C5 (Counting)-focused data:** We can increase counting data under multi-view and occlusion-heavy scenes to force MLLMs’ discrete numerosity reasoning.
> >
> > **(2) C3 (Spatial Relationship) + C10 (Situated Observation)-focused data:** Spatial Relationship is one of the most fundamental yet frequently misunderstood abilities. Compared with standard datasets that describe relations in an abstract way, capability-aware data can make the visual spatial relationships explicit.
> >
> > For example, instead of saying “the table is to the left of the sofa”, we can specify:
> > “When facing the front of the sofa, the table is on its left.”
> > This forces the model to learn what visual evidence actually determines left, right, front, or back, since changing the viewpoint will also change the perceived spatial relationship. Such data can help models acquire robust and flexible visual-spatial understanding, rather than relying on memorized biases or textual shortcuts.
> >
> > **(3) C7 (Multi-view Fusion) + C8 (Forward Thinking) + C9 (Reverse Reasoning):** We can deliberately construct more long-horizon and reasoning-heavy spatial cases, where the model must integrate information from multiple views and reason about possible routes or outcomes. Such data naturally strengthens planning-oriented spatial capabilities.
> >
> > Such capability-structured corpora would help models learn spatial capabilities in a decomposed and interpretable way, rather than relying only on large-scale correlations that emerge from scaling.
> >
> > We expect this paradigm to bring several benefits: (1) it can significantly improve generalization on downstream spatial tasks; (2) reduce data cost, because targeted spatial data is more informative than generic image–text pairs; (3) and increase training transparency and controllability, since each sample directly corresponds to explicit spatial abilities.
> >
> >  In this sense, SpaCE-10 serves not only as an evaluation benchmark but also as a foundation for designing capability-aligned training curricula.
> >
> > **2. Capability-driven task decomposition for embodied applications:**
> >
> > Building on the capability-aware training paradigm described above, models trained with structured spatial abilities are more likely to develop **task-decomposition ability**, which are essential for real embodied systems. In many embodied applications, MLLMs with this ability can effectively serves as the system’s **embodied spatial brain**. It is not only answering questions, but helping the robot decompose, interpret, and execute full spatial tasks.
> >
> > For example, in a typical home-robot find-and-retrieve instruction such as
> > **“Find the watch near the nightstand and bring it to me”**, the internal reasoning process naturally involves multiple atomic capabilities:
> >
> > **C1**: recognizing the watch and nightstand to establish task-relevant objects;
> >
> > **C3 + C10**: resolving their relative spatial relationship from the robot’s current viewpoint;
> >
> > **C7**: integrating observations across viewpoints as the robot moves;
> >
> > **C8/C9**: planning, verifying, and adjusting feasible paths and actions.
> >
> > Therefore, the capability system in SpaCE-10 also provides the foundations for **task decomposition** and **cross-scene spatial generalization** in real-world applications.

---

> ### Author Response · Authors · 2025-11-23
> **Response to sqZm-3**
>
> **Q3:** Please provide a detailed breakdown of the SpaCE-10 benchmark. Specifically, what is the total number of QA pairs, the number of distinct images used, and the precise distribution of samples across the 10 atomic and 8 compositional capabilities?
>
> **A3:** We thank the reviewer for this suggestion to provide a more detailed breakdown of SpaCE-10.
> Below, we summarize the exact statistics. SpaCE-10 contains **5,122 QA pairs** in total (4,152 single-answer and 970 multiple-answer), covering **811 distinct indoor scenes** with **6,488 multi-view images**. The eight QA types contain the following number of samples:
> **EQ: 388, SQ: 645, SA: 785, OO: 809, OS: 654, EP: 1,013, FR: 800, SP: 80.**
>
> Using the capability–task mapping defined in Table 5, we compute the precise distribution of samples across the ten atomic spatial capabilities:
> **C1: 5,122,
> C2: 4,477,
> C3: 2,343,
> C4: 785,
> C5: 1,033,
> C6: 2,230,
> C7: 5,122,
> C8: 5,122,
> C9: 1,893,
> C10: 2,343.**
>
> Additionally, the total number of QAs and the C1-C10 capability coverage are shown in Fig. 2 of the main paper, and the capability-QAs mapping is provided in Tab. 5 of the appendix. In the revision, we will additionally include these detailed numerical statistics in the appendix for completeness and reproducibility.
>
> We thank the reviewer again for highlighting this, and we have updated benchmark statistics in the revision.
>
>
>
>
>
> ### **Reference:**
>
> [1]nuScenes: A multimodal dataset for autonomous driving.
>
> [2]Argoverse 2: Next Generation Datasets for Self-Driving Perception and Forecasting.
>
> [3]ViNT: A Foundation Model for Visual Navigation.
>
> [4]CoNVOI: Context-aware Navigation using Vision Language Models in Outdoor and Indoor Environments.

---

> > ### Comment · Reviewer_sqZm · 2025-11-25
> >
> > Thank you for your reply, and I really appreciate the insights shared in response to the second weakness. I will stay positive in the remaining rebuttal and discussion period. Hope to see this paper accepted.

---

> > > ### Author Response · Authors · 2025-11-25
> > >
> > > Thank you very much for your consistently supportive feedback！

---

### Official Review · Reviewer_ffEP · 2025-11-03

**Soundness:** 3
**Presentation:** 3
**Contribution:** 3
**Rating:** 8
**Confidence:** 3

**Summary:**

This paper introduces SpaCE-10, a large-scale benchmark designed to evaluate the compositional spatial intelligence (CSI) of multimodal large language models (MLLMs). The authors argue that existing benchmarks fail to disentangle and systematically assess spatial capabilities in MLLMs. SpaCE-10 defines 10 atomic spatial capabilities (e.g., object recognition, counting, spatial relationship, multi-view fusion) and combines them into 8 compositional QA tasks covering both perception and reasoning.
The benchmark is built upon 811 real indoor 3D scenes collected from four public datasets, producing over 5,000 high-quality question-answer pairs through a hierarchical semi-automated annotation pipeline involving both GPT-4o and human experts.
The authors evaluate ~50 open- and closed-source MLLMs, including GPT-5, LLaVA-OneVision, Qwen2.5-VL, and InternVL3.5. Results show that even the best models perform far below human level (≈53% vs. 91%), with major deficiencies in counting, reverse reasoning, and multi-capability composition. The authors release the dataset and code to promote further research in spatial reasoning for MLLMs.

**Strengths:**

- The paper convincingly identifies the lack of a unified benchmark for compositional spatial intelligence, distinguishing SpaCE-10 from prior 2D/3D QA datasets.
- The hierarchical annotation pipeline (combining automated generation and human validation) is well thought-out, ensuring both scalability and quality.
- By defining 10 atomic spatial capabilities and mapping them to 8 compositional QA types, the benchmark enables fine-grained capability diagnosis rather than raw accuracy comparison.
- The inclusion of nearly 50 MLLMs, spanning 2D and 3D models from 1B to 241B parameters, provides an impressive empirical scope and meaningful comparative insights.

**Weaknesses:**

- Although human validation is used, the quality and potential bias of GPT-generated QAs could influence benchmark reliability; the paper could analyze this more rigorously.
- SpaCE-10 focuses on indoor 3D environments; its applicability to outdoor or dynamic (temporal) spatial reasoning remains unexplored.
- While overall accuracy drops are discussed, the paper could provide deeper qualitative examples showing failure modes and reasoning errors.

**Questions:**

- How does SpaCE-10 ensure that the compositional QA pairs do not contain linguistic or visual shortcuts that allow models to guess without full reasoning?
- Could the benchmark be extended to temporal or embodied spatial reasoning, e.g., predicting next-step navigation or multi-step manipulation tasks?
- Given that GPT-4o is used in both generation and evaluation, how is data leakage or model bias avoided?

---

> ### Author Response · Authors · 2025-11-23
> **Response to ffEP**
>
> We sincerely thank you for your positive and encouraging review. Thank you for the time and care you dedicated to evaluating our work. Below are our responses to your comments and questions.
>
> **Q1:** Although human validation is used, the quality and potential bias of GPT-generated QAs could influence benchmark reliability; the paper could analyze this more rigorously.
>
> **A1:** We thank the reviewer for raising this insightful question. Below, we clarify why any potential bias from GPT-assisted items is limited and well-controlled in SpaCE-10.
>
> **1. Multiple safeguards in the QA construction pipeline**
>
> (1) **Structured data preparation before QA generation.** Before generating any QA, we build a verifiable scene structure: select multi-view keyframes, generate and inspect-refine 2D/3D captions, and apply a rule-based extractor to obtain object/attribute/relation descriptions. Basic QAs are generated after this step, so all questions and options stay tied to observable facts by these structured resources rather than free-form text.
>
> (2) **Hybrid Basic-QA generation.** QAs come from a mixture of template-based items (Entity Quantification), human-written items (Spatial Planning), and LLM-assisted items (for the remaining types). LLM use follows fixed templates, well-designed prompts, and structured data, which constrain generations and mitigate artifacts.
>
> (3) **Controlled cross-capability integration.** We compose from Basic QAs to multi-capability items via template-based integration (e.g., adding C7 (Multi-view Fusion)/C9 (Reverse Reasoning)/C10 (Situated Observation) as appropriate), which preserves traceability and minimizes bias during compositional construction.
>
> **2. Efficient human validation with a dedicated UI**
>
> We further apply expert validation: 3 experts curate multi-view snapshots (>38 hours); a dedicated validation UI enables 3 experts to review and delete any item with incorrect options/answers (>112 hours). In total, expert effort exceeds 150 hours.
>
> Therefore, in SpaCE-10, potential bias is effectively controlled. The consistent results in the main paper (e.g., performance comparisons and capability analyses across nearly 50 MLLMs) can also validate it. Thanks for your insightful question again, and we hope this addresses your concern.

---

> ### Author Response · Authors · 2025-11-23
> **Response to ffEP-2**
>
> **Q2:** SpaCE-10 focuses on indoor 3D environments; its applicability to outdoor or dynamic (temporal) spatial reasoning remains unexplored. Could the benchmark be extended to temporal or embodied spatial reasoning, e.g., predicting next-step navigation or multi-step manipulation tasks?
>
> **A2:** We sincerely thank the reviewer for the thoughtful and constructive suggestion. We fully agree that expanding beyond household indoor scenes is important for advancing spatial intelligence. In particular, outdoor environments are central to open-world spatial intelligence (e.g., navigation, locomotion, and large-scale spatial planning), and we plan concrete extensions in the next iteration of SpaCE-10.
>
> We conduct an initial survey and identify large-scale multi-sensor autonomous-driving datasets (e.g., nuScenes [1] and Argoverse 2.0 [2]) that provide synchronized LiDAR, multi-view RGB, and HD maps. These resources naturally support capability-centric evaluations such as multi-view fusion under open-world perspectives and spatial localization with map priors in both 2D and 3D. For embodied outdoor navigation, models/datasets such as ViNT [3] and CoNVOI [4] indicate feasible paths to probe long-horizon, goal-conditioned reasoning beyond indoor settings.
>
> We also plan a temporal extension to address applicability to dynamic scenes. Concretely, we can evaluate online spatio-temporal reasoning by testing frame-to-frame persistence and identity tracking under occlusion, motion-aware relational reasoning over time (e.g., which path remains collision-free in the next *k* frames). We can also take spatial works like OST-Bench [5] as a starting point for temporal task guidance and adapt SpaCE-10’s design to develop temporal atomic spatial capabilities.

---

> ### Author Response · Authors · 2025-11-23
> **Response to ffEP-3**
>
> **Q3:** While overall accuracy drops are discussed, the paper could provide deeper qualitative examples showing failure modes and reasoning errors.
>
> **A3:** We sincerely thank the reviewer for the thoughtful comments about failure modes. In response, we conducted a focused analysis across all eight QA types, examining how different atomic capabilities from C1 to C10 contribute to the observed errors. These visual examples shown in Fig.9-17 reveal a wide range of limitations, from basic perception issues to more complex challenges such as multi-view fusion, functional reasoning, and situated spatial understanding. Together, they confirm that current MLLMs still lack reliable mastery of many core spatial abilities, further underscoring the necessity of a capability-centered benchmark like SpaCE-10. For clear demonstrations, we include the full detailed analysis in the updated Appendix I.1 section.
>
> **Q4:** How does SpaCE-10 ensure that the compositional QA pairs do not contain linguistic or visual shortcuts that allow models to guess without full reasoning?
>
> | Num |   EQ  |  SQ  |   SA  |   OO  |   OS  |   EP  |   FR  |   SP  | Overall |
> |-----|-------|------|-------|-------|-------|-------|-------|-------|---------|
> | Random  | 25.3  | 25.1 |  24.9 | 23.1  |24.0  |25.0 | 23.8  | 26.3  |24.7
> | Non-Visual   | 28.3  | 25.6 |24.5  |25.9  | 25.0  | 25.2  | 23.6  | 25.0 | 25.4 |
> | Visual | 36.6  | 29.5 | 42.9  | 51.7  | 34.5  | 26.6  | 60.6  | 37.5  | 40.0 |
>
> We thank the reviewer for the question. To verify that SpaCE-10 does not contain linguistic or visual shortcuts, we evaluate three baselines using InternVL3-8B [6].
>
> (1) A random guesser achieves an overall accuracy of 24.7%, matching chance level.
>
> (2) The non-visual (language-only) variant of InternVL3-8B performs similarly with 25.4%, indicating that the benchmark does not contain linguistic shortcuts that allow models to answer without visual grounding.
>
> (3) The full visual InternVL3-8B achieves a substantially higher 40.0%, showing that true visual and spatial reasoning is required to solve SpaCE-10.
>
> Additionally, as discussed in Q3, we provide full qualitative analyses of model failure modes in Appendix I.1, further confirming that models succeed only when they reason over visual evidence rather than exploiting shortcuts.
>
> **Q5:** Given that GPT-4o is used in both generation and evaluation, how is data leakage or model bias avoided?
>
> **A5:** We thank the reviewer for raising this important concern. We clarify below how SpaCE-10 avoids data leakage and model bias, even though GPT-4o [7] is involved in both generation and evaluation.
>
> **1. GPT-4o is not used to judge correctness.**
>
> GPT-4o is only used in evaluation as a lightweight post-processor when a model fails to follow the instruction format (e.g., it outputs free-form text instead of A/B/C/D). This is the same mechanism used in MMBench [8], and GPT-4o never determines whether an answer is correct.
>
> **2. GPT-4o activation frequency is extremely low.**
>
> We counted how often GPT-4o is triggered during a full evaluation of InternVL3-8B. Out of 4152 QA evaluations, GPT-4o was used only 35 times (0.84%), meaning over 99% of answers are evaluated without GPT-4o involvement. This ensures that GPT-4o has a negligible impact on final scores.
>
> **3. No evidence of bias in cross-model results.**
>
> Across nearly 50 evaluated MLLMs, performance trends are consistent and align with capability-level analyses, indicating that no systematic bias or leakage is introduced by GPT-4o.
>
> Overall, GPT-4o is used only as a rare fallback for output formatting, never for correctness, and is strictly controlled. Combined with our MCQ evaluation protocol, this ensures that data leakage and model bias are effectively avoided.
>
> ### **Reference:**
> [1] nuScenes: A multimodal dataset for autonomous driving.
>
> [2] Argoverse 2: Next Generation Datasets for Self-Driving Perception and Forecasting.
>
> [3] ViNT: A Foundation Model for Visual Navigation.
>
> [4] CoNVOI: Context-aware Navigation using Vision Language Models in Outdoor and Indoor Environments.
>
> [5] OST-Bench: Evaluating the Capabilities of MLLMs in Online Spatio-temporal Scene Understanding.
>
> [6] InternVL3: Exploring Advanced Training and Test-Time Recipes for Open-Source Multimodal Models.
>
> [7] GPT-4 Technical Report.
>
> [8] MMBench: Is Your Multi-modal Model an All-around Player?

---

### Author Response · Authors · 2025-11-23
**General Response**

We sincerely thank all reviewers, AC, PC, and SPCs for your time, effort, and significant reviews. Our submission receives an overall positive average score (6), with a mix of strong recommendations and relatively reserved assessments (8, 8, 4, 4). We are also grateful for reviewer sqZm’s note indicating the willingness to assign a score of 9 if the system permitted it, reflecting the strong confidence in the contribution of SpaCE-10.

**Summary of Reviewer Positive Feedback**

1. **Clear and important contribution:** SpaCE-10 fills the capability gap of a unified benchmark for compositional spatial intelligence. (ffEP, sqZm)

2. **Well-structured capability framework:** The 10 atomic and 8 compositional capabilities provide intuitive and fine-grained diagnosis. (ffEP, sqZm)

3. **High-quality data pipeline:** The automated-plus-human annotation ensures scalable and reliable QA pairs. (ffEP, D8Pf)

4. **Extensive evaluation:** The nearly 50-model study offers broad and meaningful comparative insights. (ffEP, sqZm, D8Pf, WP4t)

5. **Clear writing and presentation:** The paper is easy to follow, well motivated, and supported by effective figures. (sqZm, WP4t)

**Summary of Key Revisions and Discussion**

1. **Future extensions to outdoor and industrial scenarios:** Clarified the motivation for focusing on indoor scenes and outlined concrete extension paths. (ffEP, sqZm, D8Pf)

2. **More qualitative analysis:** Added detailed error-pattern analysis with visualizations. (ffEP)

3. **Clarification of GPT-related bias:** Resolved the misunderstanding by emphasizing our contribution in data generation and validation. (ffEP)

4. **Performance and input view count:** Added more ablations and provided an in-depth analysis of the improvement curve. (D8Pf)

5. **Precise Numeric and Multiple-Choice Question (MCQ) outputs:** Systematically discussed the strengths and weaknesses of numeric versus MCQ formats. (D8Pf)

6. **Atomic capability co-occurrence:** Clarified why certain capabilities naturally co-occur and explained their inherent dependencies. (WP4t)

We appreciate all reviewers’ feedback and remain committed to further improving the work. The following sections address each reviewer’s comments in detail.

---

### Author Response · Authors · 2025-11-30
**Thank AC for taking over our submission and Summary of Review-Rebuttal phase**

Thank you for taking on this paper and for being invited to participate in our discussion with the reviewers. We understand that the special circumstances this year have placed an additional burden on ACs, and we genuinely appreciate your time and effort. **To help you efficiently assess our work, we provide a concise summary of our contribution, the reviewers' evaluation, and the key improvements made during rebuttal.**

**(I) Our main contribution**

(1) We introduce SpaCE-10, the first benchmark that proposes to evaluate and disentangle compositional spatial intelligence by defining an atomic spatial capability pool. This paradigm enables capability-level assessment of MLLMs and shifts the focus from coarse overall accuracy to a more fine-grained understanding of the spatial capability.

(2) We design a hierarchical semi-automated pipeline that converts structured multi-view scene descriptions into high-quality and controllable compositional QA pairs across multiple evaluation formats, including 3D inputs and multi-choice questions.

(3) We conduct large-scale evaluations of nearly 50 MLLMs, revealing overall accuracy, atomic capability performances, failure patterns, and 2D-3D model gaps, offering concrete insights for advancing spatial intelligence.

**(II) Strengths recognized by the reviewers**

(1) **Clear and important contribution (@ffEP, @sqZm):** SpaCE-10 fills the capability gap for compositional spatial intelligence and establishes a meaningful benchmark direction.

(2) **Well-designed capability framework (@ffEP, @sqZm):** The 10 atomic and 8 compositional capabilities provide intuitive, interpretable, and fine-grained spatial diagnosis.

(3) **High-quality annotation pipeline (@ffEP, @sqZm, @D8Pf):** The automated-plus-human labeling ensures scalable and reliable QA generation and validation.

(4) **Extensive evaluation (@ffEP, @sqZm, @D8Pf, @WP4t):** Nearly 50 MLLMs of diverse sizes offer strong comparative insights for the community.

(5) **Clear writing & good presentation (@sqZm, @WP4t):**  The paper is easy to follow, well-structured, and well-demonstrated.

**(III) Key improvements and clarification made during rebuttal**

(1) **Future extensions  (@ffEP, @sqZm, @D8Pf):** Justified the future extension plan on indoor domains and outlined concrete extensions to outdoor and industrial settings.

(2) **More qualitative analyses (@ffEP):** Added detailed error-pattern visualizations and discussed failure modes across capability types in Appendix I.1.

(3) **Clarification of GPT-related bias and shortcut (@ffEP):** Eliminated misunderstanding by demonstrating the usage of GPT-4o and the details of our pipeline design. Additional experiments between random choice, non-visual input (only language), and with visual input validated that there is no shortcut in SpaCE-10.

(4) **Input image number and performance variance analysis (@D8Pf):** Added more ablations and elaborated on the multi-view improvement curve.

(5) **Numeric vs MCQ outputs (@D8Pf):** Clarified the motivation of using MCQ, provided a systematic comparison between Numeric and MCQ outputs, and discussed the current stage of MLLMs in numeric spatial reasoning.

(6) **3D vs 2D MLLMs performance (@D8Pf):** Clarified the details of testing 3D models and pointed out two key gaps that need to be improved between 3D and 2D MLLMs.

(7) **Atomic capability co-occurrence (@WP4t):** Explained natural capability dependencies and clarified why some capabilities consistently appear together.

(8) **Capability difference between other methods (@WP4t):** Re-highlighted our core capability-based contributions and systematically compared with involved capabilities in existing works.

**(IV) Unanimous Reviewer Endorsement**

After the rebuttal (**before incident happened**), we received positive feedback from all reviewers:

@sqZm **(rating: remain 8 after rebuttal)**: “Thank you for your reply, and I really appreciate the insights shared in response to the second weakness. I will stay positive in the remaining rebuttal and discussion period. Hope to see this paper accepted.”

@D8Pf **(rating: 4 →  6 after rebuttal)**: “Overall, I appreciate the author’s detailed explanations and I’m willing to increase my rating to 6.”

@WP4t **(rating: 4 → 6 after rebuttal)**: “Thanks for the explanation. I will adjust the score to 6.”


**Final overall score: 8, 8, 6, 6 (avg. 7.0)**

Additionally:

@sqZm **expressed that a score of 9 would be assigned if permitted.**

The comprehensive improvements, including clearer clarification of our data pipeline and evaluation setting, expanded diagnostic analyses with additional qualitative studies, and strengthened comparisons across input formats, model types, and capability relationships, have substantially enhanced the clarity, robustness, and overall technical soundness of SpaCE-10.

We hope this summary facilitates your evaluation and remain available for any clarifications.

Best regards,

The Authors

---

### Meta-Review · Area_Chair_d5iT · 2026-01-05

**Summary:**

All reviewers recognize the value of a unified benchmark for compositional spatial intelligence. The proposed benchmark fills a blank where existing spatial reasoning benchmark fail to cover. The hierarchical taxonomy of spatial reasoning ability used for organization enables in-depth analysis of MLLMs. Extensive evaluation across diverse MLLMs offers meaningful comparative insights.

**Reviewer Concerns:**

Authors provide comments for all concerns raised by reviewers. Most of them are well addressed.

**Reviewer ffEP**'s W1 questions about the bias of GPT-generated QAs in this benchmark. Authors provide detailed feedback about how they control the bias through carefully designed data production pipeline. However, the expected analysis about the potential bias mentioned by reviewer is missing.

**Reviewer Scores:**

For **Reviewer ffEP (score 8)** and **Reviewer sqZm (score 8)**, I think their score will probably remain accept.

For **Reviewer WP4t (score 4)** and **Reviewer D8Pf (score 4)**, their scores are likely to increase to weak accept or accept.

---

### Decision · Program_Chairs · 2026-01-26

Accept (Poster)